# In situ cryo-electron tomography reveals the asymmetric architecture of mammalian sperm axonemes

Zhen Chen[1,2], Garrett A. Greenan[1,2], Momoko Shiozaki[3], Yanxin Liu[2], Will M. Skinner[4], Xiaowei Zhao[3], Shumei Zhao[3], Rui Yan[3], Zhiheng Yu[3], Polina V. Lishko[4,5], David A. Agard[2] ✉ & Ronald D. Vale[1,3] ✉

The flagella of mammalian sperm display non-planar, asymmetric beating, in contrast to the planar, symmetric beating of flagella from sea urchin sperm and unicellular organisms. The molecular basis of this difference is unclear. Here, we perform in situ cryo-electron tomography of mouse and human sperm, providing the highest-resolution structural information to date. Our subtomogram averages reveal mammalian sperm-specific protein complexes within the microtubules, the radial spokes and nexin–dynein regulatory complexes. The locations and structures of these complexes suggest potential roles in enhancing the mechanical strength of mammalian sperm axonemes and regulating dynein-based axonemal bending. Intriguingly, we find that each of the nine outer microtubule doublets is decorated with a distinct combination of sperm-specific complexes. We propose that this asymmetric distribution of proteins differentially regulates the sliding of each microtubule doublet and may underlie the asymmetric beating of mammalian sperm.

Eukaryotic flagella and motile cilia are whip-like organelles, the rhythmic beating of which propels unicellular eukaryotes through fluids, clears dust particles in respiratory tracts and enables the swimming of sperm cells of various species[1–3]. Most flagella from protozoa to mammals share a conserved core structure, the axoneme, composed of nine doublet microtubules (doublets) arranged in a circle around a central pair complex of two singlet microtubules (the 9 + 2 configuration, Fig. 1a)[4,5]. Dyneins, microtubule-based molecular motors anchored on the nine doublets, drive the relative sliding of neighboring doublets[6,7]. However, if all dyneins were active at once, forces around the circle of the nine outer doublets would be canceled and the axoneme would not bend[5,8]. To produce rhythmic beating motions, non-motor protein complexes are required to regulate dynein activities across the axoneme structure[5,9–15]. The largest and most critical of these regulatory

complexes are the radial spokes (RSs) that bridge the outer doublets to the central pair complex and the nexin–dynein regulatory complexes (N-DRCs) that cross-link neighboring doublets and regulate dynein activities across the axoneme structure.

Flagella from different cells display a wide variety of beating patterns, from the planar and symmetric waveforms observed in flagella of unicellular organisms and sea urchin sperm, to the various non-planar and asymmetric waveforms displayed by different mammalian sperm[8]. The structural and regulatory mechanisms underlying these different waveforms are poorly understood. Much of our current structural understanding of axonemes is derived from studies of *Chlamydomonas* and sea urchin sperm flagella using advanced cryo-electron tomography (cryo-ET) and image processing[16–18]. Apart from minor variations of the dyneins on a subset of the nine doublets, most of the other motor

[1]Department of Cellular and Molecular Pharmacology, University of California, San Francisco, San Francisco, CA, USA. [2]Department of Biochemistry and Biophysics, University of California, San Francisco, San Francisco, CA, USA. [3]Janelia Research Campus, Howard Hughes Medical Institute, Ashburn, VA, USA. [4]Department of Molecular and Cell Biology, University of California, Berkeley, CA, USA. [5]Center for Reproductive Longevity and Equality, Buck Institute for Research on Aging, Novato, CA, USA. ✉e-mail: david@agard.ucsf.edu; valer@janelia.hhmi.org

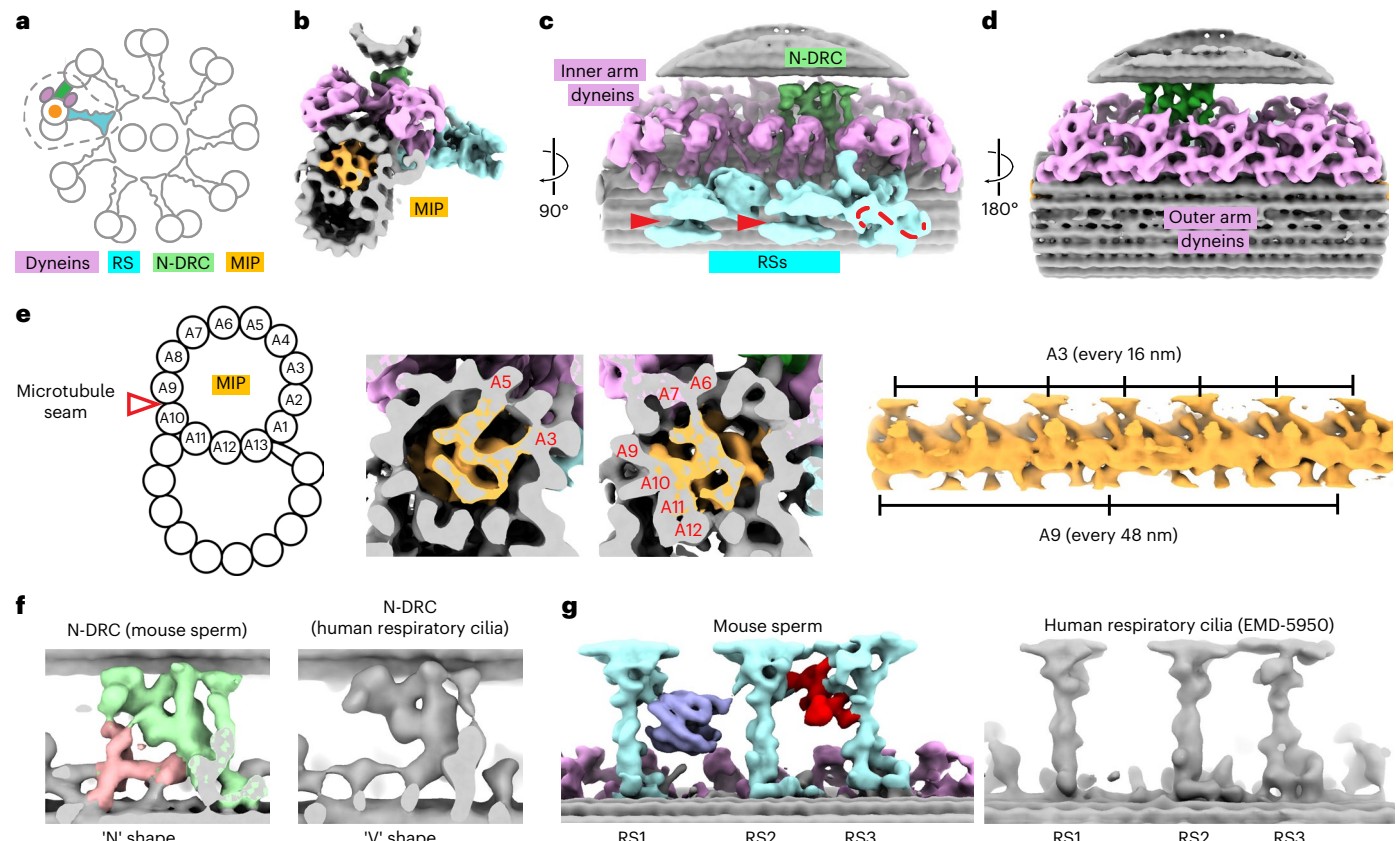

**Fig. 1 | The consensus average of nine doublets in mouse sperm possesses unique features in non-motor protein complexes including MIPs, N-DRCs and RSs. a**, Schematic of a cross-section view of the conserved (9 + 2) configuration of axonemes in motile cilia. One doublet is highlighted, and its associated motor and non-motor protein complexes are labeled. The dyneins, N-DRCs, MIPs and RSs are colored in pink, green, orange and cyan, respectively. **b–d**, Three views of the consensus subvolume average of nine doublets in mouse sperm axonemes. Different protein complexes are highlighted as described in **a**. **c**, The clefts in the top of RS1 and RS2 and the 'S'-shaped head of RS3 are indicated by the red arrowheads and dashed line, respectively. **e**, Schematic of the doublet microtubule with individual protofilaments labeled. Two cross-sections of the A tubule and one longitudinal view of the isolated MIP densities are shown. The MIPs are colored in orange. The protofilaments connecting to the MIPs and the periodicities for the connections are indicated. **f,g**, Comparison of densities of N-DRCs and RSs from mouse sperm flagella (this study) and human respiratory cilia (EMD-5950)[31]. **f**, Additional densities in the mouse sperm N-DRCs are highlighted (light red). **g**, The unique densities of a barrel and an RS2–RS3 cross-linker in the mouse sperm axoneme are highlighted in blue and red, respectively.

and non-motor protein complexes were found to be the same across the nine outer doublets. A unique bridge-like structure that cross-links two neighboring doublets is proposed to constrain the plane of bending[16–19]. The pseudo-ninefold symmetry and the bridge structure are thought to be important for generating equivalent beating amplitudes in the opposite directions, leading to planar and symmetric waveforms.

Comparable structural information for mammalian sperm, which display varied non-planar asymmetric waveforms[20–22], has lagged behind. A technical challenge in using modern cryo-electron microscopy to investigate mammalian sperm flagella is their thickness (>500 nm), which is close to the upper limit for the widely used 300-kV transmission electron microscopes (TEMs). Recently, cryogenic focused ion beam–scanning electron microscopy (cryo-FIB–SEM) and cryo-ET have been applied to study in situ macromolecular structures in sperm axonemes from mammals[23]. However, the limited data obtained in the previous study precluded processing strategies to analyze individual microtubule structures within the axonemes and also their spatial relationships in situ.

Here, we combined cryo-FIB–SEM and in situ cryo-ET with data-processing strategies to study the contextual assembly of different microtubule-based structures within mouse and human sperm flagella. Our data provide the highest-resolution information to date for mammalian sperm axonemes. Furthermore, our data reveal non-motor protein complexes in mammalian sperm that are not

found in axonemes of other mammalian cilia and non-mammalian sperm. We show that each of the nine outer doublets is unique with regard to the composition of regulatory complexes including RSs and N-DRCs. The distribution of regulators varies between mouse and human sperm. We propose that the asymmetric distribution of these regulatory complexes across the axoneme could contribute to the asymmetric and non-planar beating waveforms of various mammalian sperm.

## Results

### Sperm-specific features revealed by subvolume averaging

Freshly extracted mouse sperm were vitrified on electron microscopy (EM) grids and loaded into a cryo-FIB–SEM device to generate lamellae of ~300 nm in thickness (Extended Data Fig. 1a,b). The lamellae were then imaged using a Krios 300-kV TEM, and dose-symmetric tomographic tilt series (±48°) around the axoneme were then acquired (Extended Data Fig. 1c). Our images showed detailed molecular features including the double-bilayer membranes of the surrounding mitochondria and individual microtubule protofilaments (Extended Data Fig. 1e,f). Three-dimensional (3D) tomograms were reconstructed from the tilt series, which revealed repetitive axonemal dyneins and RSs along the outer doublets as well as periodic protrusions from the singlet microtubules of the central pair complex (Extended Data Fig. 1h,i). The periodicities of the RSs and central pair protrusions are ~96 nm

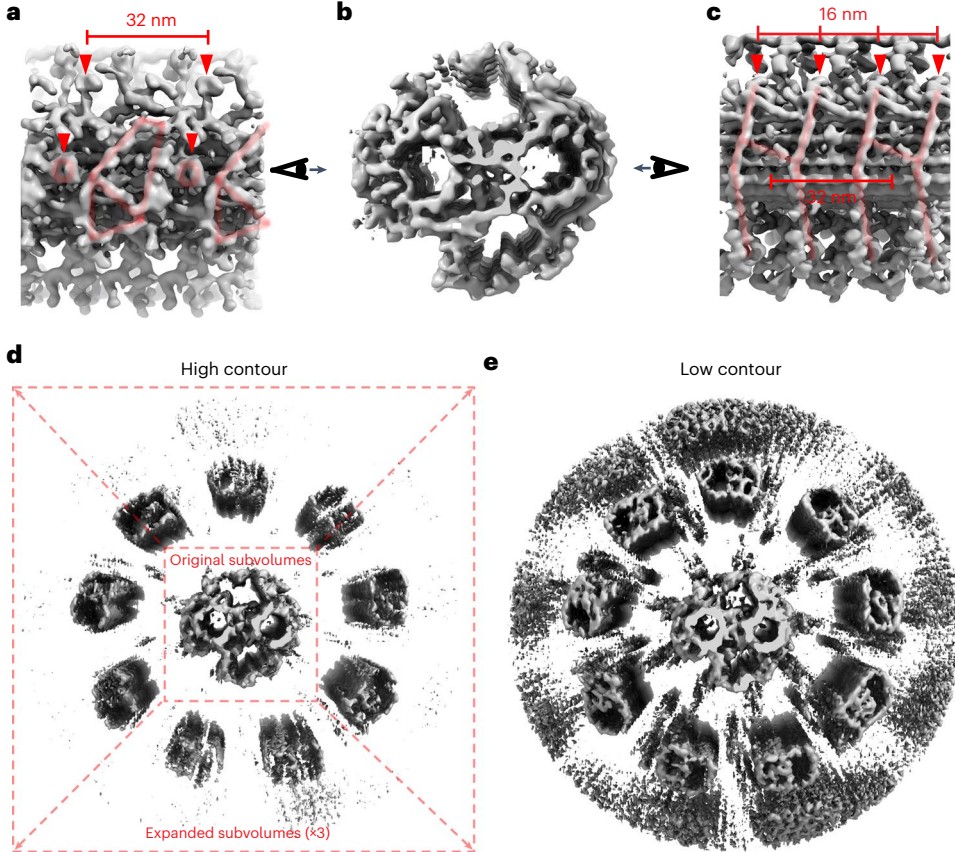

**Fig. 2 | The central pair complex presents asymmetric surfaces in different directions. a–c**, Three views of the central pair complex of the mouse sperm axoneme. **a,c**, Periodic protrusions are indicated by red arrowheads and highlighted by light red lines, and their periodicities are labeled. **d**, An average of the entire axoneme was calculated by expanding the original subvolumes for the central pair complex threefold and averaging without further alignment. At high contour, densities of nine doublet microtubules are resolved at nine distinct radial positions. **e**, The same average as in **d** is shown at low contour. Note that the A tubules from doublet microtubules are distinguishable due to the presence of extensive MIP densities. Smeared densities corresponding to where dyneins, RSs and N-DRCs are located could be observed at lower occupancies than those for the doublet microtubules.

and ~32 nm, respectively (Extended Data Fig. 1i), consistent with those described in *Chlamydomonas* and sea urchin sperm[24,25].

To overcome the low signal-to-noise ratio of raw cryo-ET data, subvolume averaging of 96-nm-repeating units was used to reconstruct consensus density maps (Extended Data Fig. 2a). Our consensus maps of periodic units from all nine doublets revealed robust signals for individual microtubule protofilaments and other associated protein complexes that repeat every 96 nm (24 Å at Fourier Shell Correlation (FSC) = 0.143; $n$ = 9,055 subvolumes; $N$ = 69 tomograms) (Fig. 1b–d and Supplementary Figs. 1 and 2).

Inside the A tubule of the outer doublet, we observed a filamentous density of microtubule inner proteins (MIPs) that is very similar to but more extensive than that recently assigned as Tektin filaments in bovine trachea cilia[26] (Fig. 1b and Extended Data Fig. 3). The densities of MIPs have a periodicity of 48 nm, consistent with that of previously studied *Chlamydomonas* flagella and bovine trachea cilia[26,27]. We thus calculated a subvolume average of the 48-nm-repeating doublets, focusing on the microtubule only (18 Å at FSC = 0.143; $n$ = 18,153 subvolumes; $N$ = 69 tomograms) (Extended Data Fig. 3). The filamentous components have connections to 12 protofilaments of the A tubule in mouse sperm axonemes (except A4), in contrast to the more limited connections of the Tektin filaments to the A9–A13 and A1 protofilaments observed in bovine tracheal cilia (Fig. 1e and comparisons in Extended Data Fig. 3b). We observed three different modes of interaction between the MIPs and the lumen of microtubules: (1) interaction with tubulins within a single protofilament, (2) connections to the inter-protofilament space

across two neighboring protofilaments and (3) connections spanning multiple protofilaments (Fig. 1e and Extended Data Fig. 3a,b). Notably, the A9–A10 junction is where the microtubule seam of the A tubule is located[28], and we observed several sperm-specific densities spanning protofilaments A9 and A10 that are absent in the map of bovine respiratory cilia (Extended Data Fig. 3b). In addition, we observed striated densities along the helical pitch of the microtubule inside the B tubule (Extended Data Fig. 3b,c). These striations are separated by 8 nm and cover the intradimeric interface between the α and β tubulins. Together, our averages revealed sperm-specific MIPs that form an extensive interaction network inside the doublets.

Our consensus map of the mouse sperm axoneme reveals outer and inner arm dyneins similar to those observed in sea urchin sperm (Fig. 1b–d and Extended Data Fig. 4a,b). However, we observed several unique non-motor protein complexes in the mouse sperm axoneme that do not have equivalent counterparts in the reported structures from *Chlamydomonas*, *Tetrahymena* and human respiratory cilia (Fig. 1f,g)[17,24,29–31]. While the N-DRC in human respiratory cilia has a 'V' shape[31], our consensus map reveals extra densities that extend to the microtubule surface, creating a square-shaped structure (Fig. 1f). The RSs are comprised of three tower-like densities, with two adopting similar morphology (RS1 and RS2) and a third, distinct RS3 (Fig. 1b,c,g). When viewed from the 'head' of the towers, RS1 and RS2 both exhibit a cleft and a C2 symmetry axis that extends through the 'tower', while RS3 has a distinctive S-shaped surface with two holes (Fig. 1c), similar to the ones observed in human respiratory cilia[31]. By contrast, the heads of

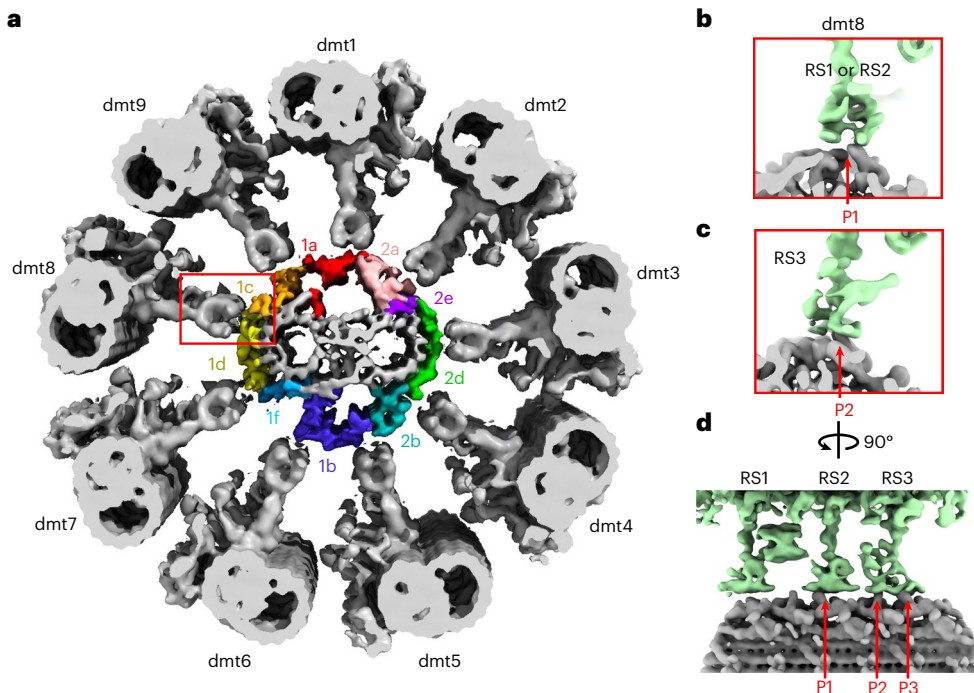

**Fig. 3 | The diverse interfaces between RSs and the central pair. a**, The arrangement of the central pair complex and nine doublet microtubules (dmt1-9) is based on averaging relative positions deduced by multibody refinement. The different protrusions of the central pair complex are colored in accordance with data from Carbajal-Gonzalez et al.[25]. The interface between the RSs from doublet 8 and the central pair complex is indicated (red rectangle). **b–d**, Three different views of the interface between the RSs from doublet 8 and the protrusions from the central pair complexes (P1, P2 and P3).

RS1 and RS2 from *Chlamydomonas* and *Tetrahymena* do not have the deep cleft[13–15,29,30], the RS3 stump from *Chlamydomonas* is much shorter, and RS3 from *Tetrahymena* has a smaller surface area of the head[15,29] (Extended Data Fig. 4c). Multiple additional densities, not found in respiratory cilia or unicellular organisms, were observed between the three spokes in the mouse sperm axoneme (Fig. 1g). First, we observed an ~20-nm-sized barrel-shaped density between RS1 and RS2, consistent with extra densities in sperm axonemes reported recently[23,32]. Our higher-resolution map revealed that the barrel is composed of ten rod-shaped strands arranged in a right-handed twist configuration (Supplementary Video 1). Furthermore, densities were found to cross-link RS2 and RS3, hereafter named the 'RS2–RS3 cross-linker' (Fig. 1g). Of note, all these extra densities in the MIPs, N-DRCs and RSs are apparent even when our maps were low-pass filtered to 50 Å, a resolution lower than that of the published map of the human respiratory cilia axoneme that does not possess these features (Extended Data Fig. 4d–f). Therefore, the additional densities found in mouse sperm are not due to higher resolution in this work but most likely reflect the presence of additional sperm-specific proteins.

### The outer doublets are arranged in fixed radial positions

We next calculated a subvolume average for the 32-nm-repeating units of the central pair complex (26 Å at FSC = 0.143; $n$ = 3,062 subvolumes; $N$ = 69 tomograms) (Fig. 2a–c, Extended Data Fig. 2b and Supplementary Figs. 1 and 2). Individual protofilaments of the two singlet microtubules were clearly resolved (Fig. 2b). Various proteins protrude from both microtubules, giving rise to an asymmetric cross-section contour of the central pair complex (Fig. 2b). We observed two distinct sets of MIPs inside the two singlet microtubules, both of which repeat every 32 nm (Extended Data Fig. 5a). On the external surface of microtubules, we observed both 32-nm-repeating and 16-nm-repeating protrusions. Notably, compared to the central pair complex of sea urchin sperm, where MIPs were not observed, the overall shapes of the external protrusions are very similar (Extended Data Fig. 5b), while both are

different compared to the central pair complex of *Chlamydomonas* flagella (Extended Data Fig. 5c)[25,33–35]. These comparisons suggest that the central pair complex is likely conserved from invertebrate to vertebrate sperm (animal sperm) but different from the ones from unicellular protists.

To understand how the outer doublets are arranged relative to the asymmetric central pair complex, we expanded our aligned subvolumes of the central pair complex three times to include the region where the outer doublet microtubules reside and then calculated an average without further alignment (Fig. 2d,e). Although the alignment was only performed for the central pair complex in the expanded subvolumes, nine distinct doublet microtubule densities that are parallel to the singlet microtubules could be resolved, indicating a remarkably consistent radial arrangement of doublets in the axonemes. The A and B tubules of doublet microtubule were clearly distinguishable based on the stronger MIP signals in the former. By contrast, discrete external complexes such as dyneins, RSs and N-DRCs were poorly resolved. The lack of alignment along the longitudinal direction could be caused by mismatch of the 32-nm periodicity of the central pair complex and the 96-nm periodicity of the doublet microtubules (Fig. 2d,e).

To better understand the spatial relationship between the central pair complex and the outer doublets, we performed multibody refinement by treating the central pair protrusions and the doublet structure as two rigid bodies, refining them separately and remapping them back to each raw subvolume[36]. Their relative positions were then subjected to principal-component analysis (Methods and Extended Data Fig. 2c). For all nine interfaces, we always observed that the first principal component, which explains most variations (40–50%), was parallel to the longitudinal axis of axonemes (Supplementary Video 2), suggesting that the doublets and the central pair complex from different tomograms meet at different longitudinal offsets.

The multibody refinement also yielded a map with the two rigid bodies placed at their average positions, allowing us to examine how the nine doublets interact with different protrusions of the central pair

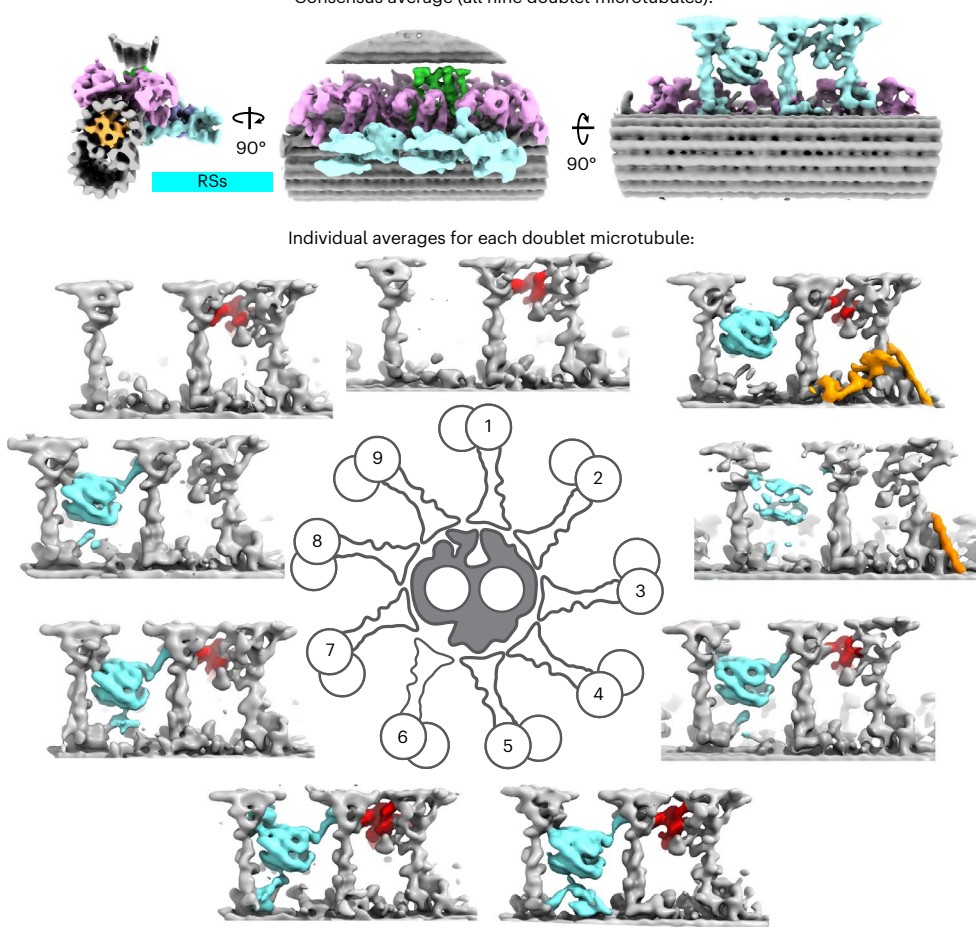

**Fig. 4 | Asymmetric distribution of sperm-specific features in RSs from the nine doublet microtubules in mouse sperm.** Three orthogonal views of the consensus average of doublets in mouse sperm are shown as in Fig. 1b–d. At the bottom, densities corresponding to RSs from the nine per-doublet averages are shown around a schematic of a cross-section view of the (9 + 2) axoneme. Common features are colored in gray, while the barrel, the RS2–RS3 cross-linker and RS3 scaffolds are highlighted in cyan, red and orange, respectively.

complex (Fig. 3a). Interestingly, we observed that most RS heads were separated by a short distance from the central pair protrusions, without any resolved densities between them (Extended Data Fig. 5d). However, at the central pair interface with doublet 8, we observed protrusions from the central pair complex fit into the 'cleft' of RS1 and RS2 and also the two holes of the 'S curve' of the head of RS3 (Fig. 3b–d). Such complementary shapes may limit the sideways movement of doublet 8 and stabilize its radial position.

**Asymmetric distribution of sperm-specific regulators**

We then sought to investigate whether the outer doublets themselves differ from each other. We grouped doublet subvolumes based on their radial positions relative to the central pair complex (numbered 1–9 as in refs. [37,38]). The subvolumes were aligned, and the averages were calculated for each of the nine doublets ($n$ = 810–954 subvolumes; $N$ = 58–64 tomograms). This processing strategy identified unique densities emerging from the inner arm dyneins of doublet 5 and connecting to the B tubule of doublet 6 (Extended Data Fig. 6). These connecting densities are similar to the '5–6 bridge' observed in the sea urchin at lower resolutions[18], validating our assignment of doublets and processing strategies. Interestingly, the RSs and other 96-nm-repeating features on both doublets 5 and 6 could be resolved concurrently after local refinement, indicating that there is a relatively consistent longitudinal offset (~20 nm) between these two doublet microtubules throughout different axonemes (Extended Data Fig. 6d). The correlation of the unique bridge densities and the consistent

offset that is observed only between doublets 5 and 6 suggest that the bridge could limit the relative sliding between this outer doublet pair (in comparison to another doublet pair in Extended Data Fig. 6g,h).

Next, we systematically compared motor and non-motor or regulatory protein complexes across the nine doublets. Outer arm dyneins across all nine doublets were indistinguishable from those of our consensus average (as shown in Fig. 1d). The densities of inner arm dyneins were also generally similar, with two exceptions. For doublet 5, densities corresponding to dyneins e and g (nomenclature defined in *Chlamydomonas*[17]) were shifted compared to those of the other doublets (Extended Data Fig. 7a), while, for doublet 9, densities for dynein b were not resolved (Extended Data Fig. 7b). These results indicate that the motor proteins are largely the same with only minor variations.

We next examined the RSs from each doublet. Strikingly, we observed that the sperm-specific features were asymmetrically distributed across the nine doublets (Fig. 4). The barrel density was not observed in doublet 1 or 9 and was present at a lower occupancy in doublet 3. In the remaining six doublets, the occupancy of the barrel was comparable to that of other repeating structures (for example, RS1). The RS2–RS3 cross-linkers are absent in doublets 3 and 8 but present in the remaining doublets. In addition, for doublets 2 and 3, we resolved extra densities close to the base of RS3 (RS3 scaffolds) that were not observed in our consensus averages or previously reported consensus averages, in which subvolumes from all nine doublets were averaged together[23,32]. This is likely because only one or two of the nine doublets possess these features (11–22%) and averaging smeared the signals.

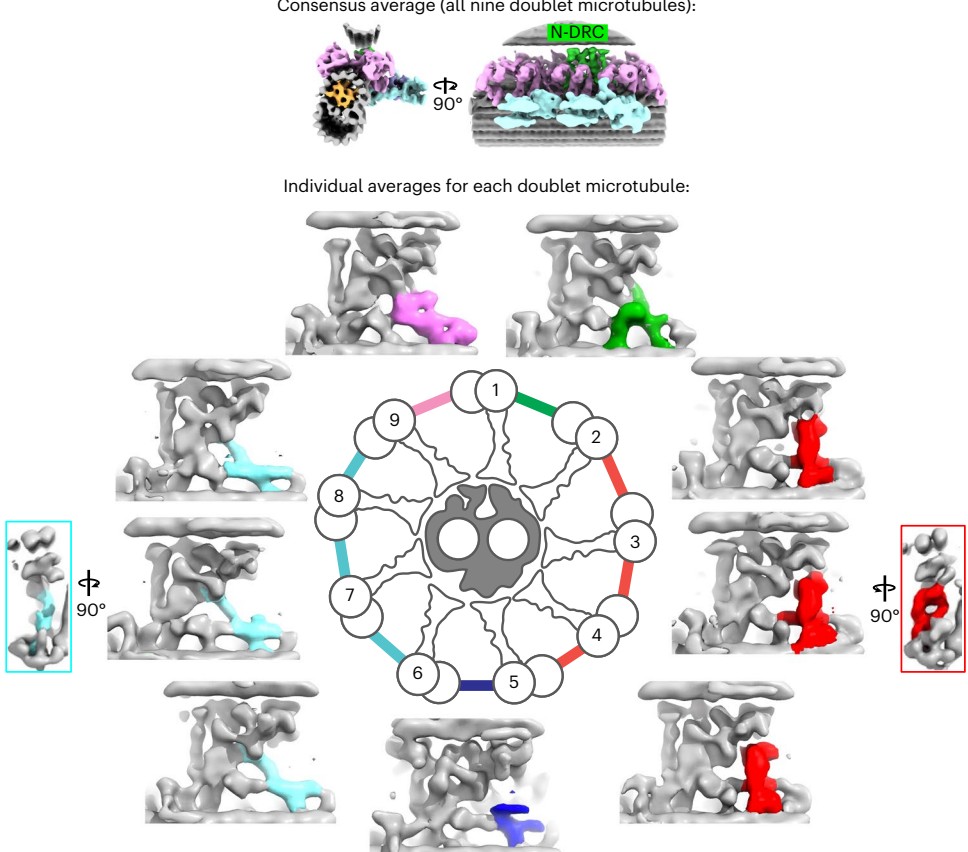

**Fig. 5 | Asymmetric distribution of sperm-specific features in N-DRCs from the nine doublet microtubules in mouse sperm.** Top, two orthogonal views of the consensus average of doublets in mouse sperm. Bottom, densities corresponding to N-DRCs from the nine per-doublet averages are shown around a schematic of a cross-section view of the (9 + 2) axonemes. Common features among the N-DRCs are colored in gray, while the unique features are highlighted.

Our processing strategy allowed us to isolate these subvolumes based on cellular contextual information, and the high occupancies of these unique structures in the respective doublets suggest their consistent presence in these specific doublets.

We also examined the N-DRCs that cross-link neighboring doublets (Fig. 5). All nine N-DRCs share the common 'V'-shape density, but the extra connections to the microtubule show heterogeneities. Interestingly, N-DRCs from doublets 2–4 share an arch-shaped density perpendicular to the microtubules, while the ones from doublets 6–8 have 45°-tilted thin strands. Doublets 1, 5 and 9 all have distinct densities, leading to five different N-DRCs. Note that all these features were observed at similar signal-to-noise levels, and because they resulted from averaging of more than 800 subvolumes sampled in ~60 different axoneme tomograms, they represent the commonly shared features within each individual doublet.

To test whether the bending states of the axonemes affect the features of RSs and N-DRCs, we curated subvolumes from tomograms with and without apparent curvatures and calculated per-doublet averages. The same set of features of RSs and N-DRCs were observed. In addition, we also collected a dataset of demembraned sperm axonemes (48 tomograms) that were not actively beating. The RSs and N-DRCs in these non-motile sperm have the same asymmetric features highlighted in Figs. 4 and 5, suggesting that the asymmetric densities were not caused by bias in macroscopic curvatures but likely reflect intrinsic compositional heterogeneities in the nine doublets.

Our in situ data also allowed us to separate axonemes of the midpiece and principal piece based on the presence of mitochondrial and fibrous sheaths, respectively. We averaged subvolumes from these two regions for each doublet and found only subtle differences in the base of RS3 of doublet 2 and also RS1 of doublet 7 (Extended Data Fig. 8a). In these per-doublet averages, we did not resolve robust densities corresponding to the connections between outer dense fibers and the respective doublets. Previous studies suggest that averages of all nine doublets in the proximal principal region from a few tomograms have such attachments[23]. However, our raw tomograms of the proximal principal piece showed that some outer dense fibers are close to the corresponding doublets and some are further away (Extended Data Fig. 8b). Such variations require per-doublet averages to be considered. However, the subset of tomograms in the proximal principal region is small, and per-doublet averaging resulted in anisotropic 3D reconstructions. An even larger dataset is required to resolve structures with such a specific and complex distribution pattern. We also calculated an average of subvolumes from the nine doublets near the beginning of axonemes (within 2 μm) and found no significant difference when compared to the overall consensus average, suggesting that the sperm-specific features are established very close to the basal region of the flagella (Extended Data Fig. 8c,d). Together, these data highlighted the overall consistency of the axoneme structure along the flagella.

Overall, our per-doublet averages showed that the distributions of various sperm-specific features for both the RS complexes and the N-DRCs follow distinct patterns, such that every doublet is decorated by a unique combination of non-motor proteins.

**A distinct asymmetric pattern in human sperm**

We next examined whether the unique outer doublet features observed in mouse sperm were also conserved in other mammalian sperm. We collected tilt series of intact human sperm without milling, focusing on the thinner principal piece of the flagellum (Extended Data Fig. 9a,b and

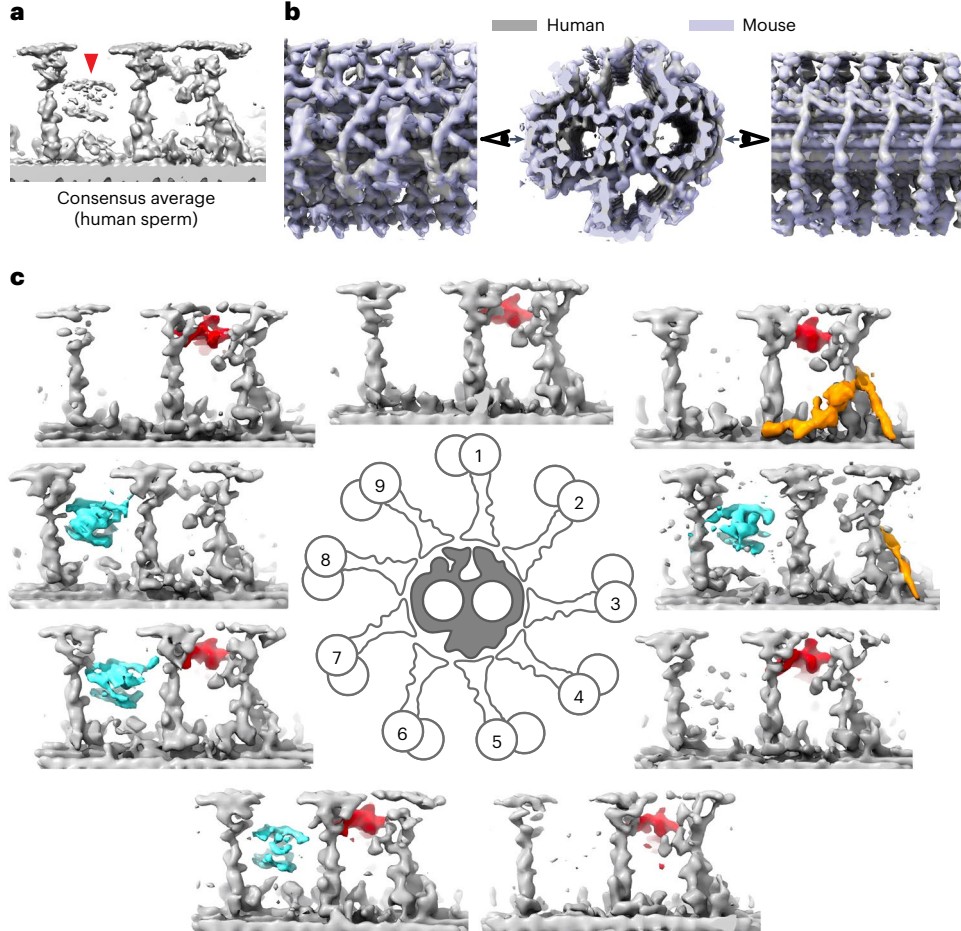

**Fig. 6 | Human sperm have a different distribution of barrels compared to that in mouse sperm. a**, A consensus average map of the RS for human sperm axonemes. The barrel density appears to have lower occupancy than the RSs (indicated by the red arrowhead), which is not the case in mouse sperm shown in Fig. 1g. **b**, The consensus average map of the central pair complex for human sperm axonemes (gray) is overlaid with one from mouse sperm axonemes (blue). **c**, Densities corresponding to RSs from the per-doublet averages are shown around a schematic of the (9 + 2) axonemes. Common features among the RSs are colored in gray, while the barrel, the RS2–RS3 cross-linker and the RS3 scaffold are highlighted in cyan, red and orange, respectively.

Supplementary Figs. 1 and 2). We then calculated consensus averages of doublet microtubules and the central pair complex (23 Å and 31 Å at FSC = 0.143; $n$ = 6,613 and 2,365 subvolumes, $N$ = 56 and 59 tomograms, respectively) (Fig. 6a,b). These consensus averages of human sperm were similar to the ones from mice, with the notable exception that the relative occupancy of the barrel between RS1 and RS2 was much lower in human sperm. Using the per-doublet processing approach described above, we then calculated averages for each of the nine doublets individually. Although the per-doublet averages from human sperm were noisier than those from mouse sperm due to greater sample thickness (>400 nm versus ~330 nm as shown in Extended Data Fig. 9c–e), they were sufficient to identify the sperm-specific features (Fig. 6c). In particular, the 5–6 bridge, RS2–RS3 cross-linkers and RS3 scaffolds show similar asymmetric distributions between human and mouse sperm. However, we observed that the barrel density only exists in four of nine doublets, in contrast to seven of nine doublets in mouse sperm axonemes (Fig. 6c). This is consistent with the lower occupancy of barrel densities in our consensus average (Fig. 6a). In particular, doublets 2–5 appear to be different in human and mouse axonemes in terms of the presence of the barrel, while the rest of the doublets are similar. We also examined the N-DRC from each doublet and found five distinct structures like those in mouse sperm (Extended Data Fig. 9f). In summary, our data indicate that the asymmetric architecture of axonemes is a general feature of mammalian sperm axonemes,

although there are intriguing variations of distribution for the barrel in human and mouse sperm.

## Discussion

Our in situ tomography studies of mouse and human sperm revealed a large ensemble of macromolecular complexes in their native cellular environment. In particular, we observed various sperm-specific features in the MIPs, RSs and N-DRCs that were not observed in mammalian respiratory cilia and non-mammalian sperm. Furthermore, we reconstructed the entire axoneme using cellular contextual information and uncovered the asymmetric architecture of the mammalian sperm axoneme, where every microtubule-based structure is decorated by a unique set of non-motor proteins (Fig. 7a,b). As these non-motor proteins regulate dynein activities based on previous studies, we propose that the asymmetries of non-motor protein complexes could modulate the sliding of the nine doublets individually to shape the species-specific asymmetric waveforms observed for mammalian sperm, as discussed below.

### Sperm-specific structures added mechanical couplings

Mammalian sperm flagella are generally much longer and wider than flagella from unicellular organisms or respiratory cilia (lengths of >45 μm versus ~10 μm and diameters of >0.5–1 μm versus ~0.3 μm). Sperm axonemes are also surrounded by additional subcellular structures,

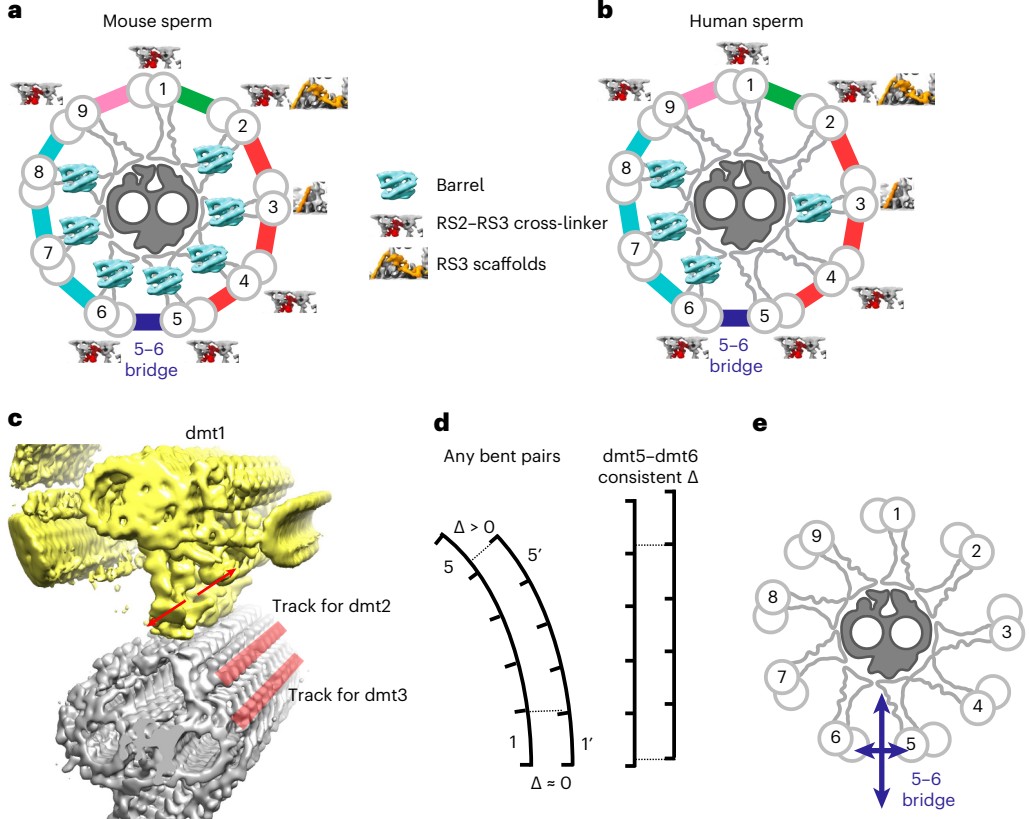

**Fig. 7 | Model: every outer doublet is surrounded by a different set of regulatory complexes in mammalian sperm.** Schematic of the (9 + 2) axonemes of mouse (**a**) and human (**b**) sperm. The doublets are numbered, and the sperm-specific regulatory complexes are labeled for each of the nine outer doublets. In particular, the components from the RSs are shown for each doublet and the N-DRCs are colored differently depending on the extra density (as in Figs. 4–6). Note that only the barrel distribution is different in mouse and human sperm axonemes. **c**, RSs from different doublets interact with specific stripes of

protrusions of the central pair complex. Our multibody refinement is consistent with the sliding hypothesis, and potential longitudinal movements of dmt1 are indicated by the two red arrows. The model is analogous to 'nine train moving on nine tracks'. **d**, Schematics showing gradual accumulations of offsets of periodic structure units between two filaments in a curved axoneme (left) and consistent offset between doublet 5–doublet 6 (right). **e**, The constant offset within tomograms and among tomograms (N = 63) would be consistent with limited horizontal bending in sperm flagella.

such as outer dense fibers and mitochondrial and fibrous sheaths, which likely present additional mechanical challenges during rhythmic beatings. Previous studies also suggested that larger bending torques are associated with mammalian sperm compared to other cilia[39], but it has remained unclear how sperm axonemes have evolved specific mechanisms to withstand the additional mechanical stress. Our averages of in situ cryo-ET revealed many additional non-motor proteins that cross-link the known axonemal components. We propose that these additional proteins function to strengthen the mechanical rigidity of the corresponding components to accommodate higher mechanical requirements of sperm axonemes.

The microtubules themselves must be able to withstand vigorous bending, and the seam is the weakest point[40,41]. Our data revealed the most extensive MIP interaction network in microtubules observed in axonemes to date. The proteins between the A9 and A10 protofilaments could stabilize the lateral interfaces within the seam. Also, the proteins that form the striations inside the B tubule could cross-link the tubulins within B2–B6 protofilaments and also couple these protofilaments laterally along the helical pitch. We also observed extensive filamentous structures within the A tubule that likely provide additional mechanical stability. Tektin-1 to tektin-4 have been known to assemble into three-helix bundles that pack along one another laterally inside the microtubule doublets of bovine tracheal cilia[26] and likely compose a part of the filamentous densities observed in sperm. Mammalian sperm also contain an additional tektin (tektin-5; Supplementary Table 1)[42],

which is a candidate for some of the additional sperm-specific densities (as shown in Extended Data Fig. 3b), although the assignment requires confirmation with higher-resolution reconstructions.

We also observed unique sperm-specific densities on the exterior of the microtubules, specifically the barrel and RS2–RS3 cross-linker between the three RSs, the RS3 scaffolds and extra densities in the N-DRC. The RSs were previously observed to tilt relative to the microtubule[43]. Additional connections between the three RSs and the scaffolds at the bases would integrate them together into a more rigid unit. Another possibility is that the coupling could lead to coordinated movement of the three RSs. The N-DRCs regulate the sliding between the neighboring doublets and prevent splaying of axonemes[12]. Extra densities linking to the microtubules could improve the stability of N-DRCs under higher mechanical stress. Together, the additional protein complexes in mammalian sperm would help to maintain the geometric integrity of the (9 + 2) microtubule configuration of the axoneme under mechanical stress during vigorous beating motions.

**The arrangement of the doublets and the central pair complex**
Previous studies suggested that the central pair complex can twist radially relative to the nine outer doublets in *Chlamydomonas* flagella[44]. Our observation of densities corresponding to the nine doublet microtubules in the average of the entire (9 + 2) axoneme can exclude the possibility of such free twisting in mammalian sperm as the doublet densities would be smeared by averaging (Fig. 2d,e). Interestingly, we observed a cleft

between the two halves of RS1 and RS2 and holes in RS3 in mammalian sperm axonemes (Fig. 1c), in contrast to the flatter surface of the RS heads from *Chlamydomonas* flagella[13–15] (Extended Data Fig. 4c). While the flat surfaces may enable twisting in *Chlamydomonas*[44], the complementary shapes of RSs and protrusions from the central pair complex in mammalian sperm may restrict such radial movements (as shown in Fig. 3b–d). Fixation of radial positions of the nine doublets also orients each of the RS complexes in proximity to unique stripes of densities of the central pair complex (Fig. 3a). This arrangement could allow functional specialization and divergent evolution of each doublet microtubule, such as the distinct sets of sperm-specific features in the nine doublets.

Our multibody refinement suggests the existence of heterogeneities in the longitudinal offsets between doublets and the central pair complex in the randomly sampled axonemes that were combined in the averages (Supplementary Video 2). This observation is consistent with the sliding hypothesis for axonemal bending, in which active dyneins generate displacement between the neighboring doublets[6]. Such movement would also lead to displacement of the doublets relative to the central pair complex along the longitudinal axis of axonemes, as though there were nine trains (doublets) moving along nine tracks (central pair protrusions) (Fig. 7c).

The 5–6 bridge is a sperm-specific feature that appears to be conserved between sea urchin[18] and mammalian sperm. Previous studies pointed out that, for two parallel inelastic microtubule filaments, bending would lead to gradual accumulation of longitudinal offsets between them if they were at different radii of the bend (Fig. 7d)[43]. As an estimate based on the reported curvature of mouse sperm[38], the offsets between neighboring doublets could differ by as much as Δ~28 nm within one 2 μm-long tomogram (details in the Methods). More importantly, the initial offset of each tomogram varies depending on how much sliding has happened upstream or downstream of the imaged area of the flagella. On the other hand, the resolved periodic features in both doublets 5 and 6 suggest that there is a consistent offset between these two doublets (Extended Data Fig. 6), not just within each tomogram but also among the 63 tomograms that contribute to the average. Due to the nature of random sampling of our imaging areas of different cells (*N* = 63 tomograms), this could happen if there is generally limited bending along the direction parallel to the plane of these two filaments (Fig. 7e). The bundling of these two doublets can create a filament stiffer than that of a single doublet, with distinct elastic properties or bending propensities in different directions. Such asymmetries of mechanical properties within the nine doublets could also contribute to asymmetric waveforms.

### Asymmetries of regulators may lead to asymmetric beatings

Our in situ data and processing strategies based on the contextual information revealed that the axoneme itself, which appeared to have 'pseudo-ninefold symmetry' in classical TEM images[3], is highly asymmetric at the molecular level. As the axoneme is the underlying engine that drives the flagellar beating motion, such asymmetries in structure could lead to asymmetric beating waveforms.

Furthermore, the asymmetries lie mostly in the mammalian sperm-specific non-motor protein complexes, including the RSs and N-DRCs. These complexes are well-established regulators of dynein motor activities, and defects in individual protein components can lead to irregular beating[9,10,12,13]. In mammalian sperm, we show here that each of the nine doublet microtubules is decorated by a unique composition of sperm-specific regulators. We speculate that the distinct molecular composition could lead to differences in the sliding speeds or bending forces for each of the doublet pairs (Fig. 7). The non-equivalent forces could lead to a deviation from the single plane of beating characteristic of more symmetric axonemes. Thus, we hypothesize that the non-uniform distribution of sperm-specific RS and N-DRC regulators are important for asymmetric and non-planar beating. Additionally, previous studies showed that human and mouse sperm have different swimming waveforms (movies in refs. [22,45]). Our

studies suggest that, although the RS barrels are conserved in human and mouse, their distribution varies and the variations could create diverse sperm swimming behaviors.

This study has revealed mammalian sperm-specific structures within the microtubules, RSs and N-DRCs that are absent in sea urchin and zebrafish sperm[46–48]. The still unknown proteins likely arose to serve functions required by natural fertilization in mammals. Sperm from different species are cast into a foreign environment and selected for their ability to reach and fertilize an egg. Sea urchin and zebrafish sperm swim freely in water, whereas mammalian sperm swim in a thin layer of viscous liquid on uneven surfaces of female reproductive tracts[49]. The viscous environment brought more mechanical challenges to the axonemes and hence the microtubule filaments inside. In addition, asymmetric and non-planar waveforms of mammalian sperm could be beneficial to navigate around 3D obstacles in reproductive tracts. Furthermore, the dimensions and physical characteristics of reproductive tracts vary among different mammals; therefore, fine-tuning the underlying molecular features to produce specialized waveforms is likely under evolutionary selection. In the future, systematic genetic and proteomic analyses of mammalian sperm-specific proteins would be valuable to connect sperm-specific axonemal structures with their functions in sperm motility and reproduction.

## Online content

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

## Methods

All chemicals were purchased from Sigma-Aldrich unless otherwise noted.

### Sample preparation

Mouse sperm were collected from 10–16-week-old C57Bl/6J mice based on a published protocol[50]. Briefly, the sperm were stripped from vasa deferentia by applying pressure to the cauda epididymis in 1× Krebs buffer (1.2 mM $KH_2PO_4$, 120 mM NaCl, 1.2 mM $MgSO_4$·$7H_2O$, 14 mM dextrose, 1.2 mM $CaCl_2$·$2H_2O$, 5 mM KCl, 25 mM $NaHCO_3$). The sperm were washed and resuspended in ~100 μl Krebs buffer for the following experiments.

Human sperm cells were collected by masturbation from healthy donors and visually inspected for normal morphology and motility before use. Spermatozoa were isolated by the swim-up procedure in HTF or HS solution as previously described[51] and then concentrated by centrifuging for 5 min at 500*g*. The supernatant was reduced to 100 μl, and the cells were resuspended.

All experimental procedures using human-derived samples were approved by the Committee on Human Research at the University of California, Berkeley, under IRB protocol number 2013-06-5395.

### Grid preparation

EM grids (Quantifoil R 2/2 Au 200 mesh) were glow discharged to be hydrophilic using an easiGlow system (Pelco). The grid was then loaded onto a Leica GP cryo plunger (pre-equilibrated to 95% relative humidity at 25 °C). The mouse sperm suspension was then mixed with 10-nm gold beads (Electron Microscopy Sciences, 25487) to achieve final concentrations of $2$–$6 × 10^6$ million cells per ml. For the demembraned mouse sperm, Triton X-100 was added to a final concentration of 0.1% in the final suspension. The human sperm suspension was mixed with 10-nm gold beads to achieve final concentrations of $0.5$–$2 × 10^6$ million cells per ml. Next, 3.5 μl of sperm mixture was added to each grid, followed by incubation for 15 s. The grids were then blotted for 4 s and plunge-frozen in liquid ethane.

### Cryogenic focused ion beam milling

Cryogenic focused ion beam milling was performed either manually using an Aquilos cryo-FIB–SEM microscope (Thermo Fisher Scientific) or automatically using an Aquilos II cryo-FIB–SEM microscope (Thermo Fisher Scientific). A panoramic SEM map of the whole grid was first taken at 377× magnification using an acceleration voltage of 5 kV with a beam current of 13 pA and a dwell time of 1 μs. Targets with appropriate thickness for milling were picked on the map. A platinum layer (~10 nm) was sputter coated, and a gas injection system was used to deposit the precursor compound trimethyl(methylcyclopentadienyl)platinum(IV). The stage was tilted to 15–20°, corresponding to a milling angle of 8–13° relative to the plane of grids. FIB milling was performed using stepwise decreasing current as the lamellae became thinner (1.0 nA–30 pA; final thickness, ~300 nm). The grids were then stored in liquid nitrogen before imaging.

### Image acquisition and tomogram reconstruction

Tilt series of mouse sperm were collected on a 300-kV Titan Krios TEM (Thermo Fisher Scientific) equipped with a high-brightness field emission gun (xFEG), a spherical aberration corrector, a Bioquantum energy filter (Gatan) and a K3 Summit detector (Gatan). The images were recorded at a nominal magnification of 19,500× in super-resolution counting mode using SerialEM[52]. After binning over 2 × 2 pixels, the calibrated pixel size was 3.53 Å on the specimen level. For each tilt series, images were acquired using a modified dose-symmetric scheme between −48° and 48° relative to the lamella with 3° steps and grouping of two images on either side (0°, 3°, 6°, −3°, −6°, 9°, 12°, −9°, −12°, 15°…). At each tilt angle, the image was recorded as movies divided into eight subframes. The total electron dose applied to a tilt series was 100 e$^-$ Å$^{-2}$. The defocus target was set to be −4 to −7 μm.

Tilt series of human sperm were recorded at a nominal magnification of 33,000× in super-resolution counting mode. After binning over 2 × 2 pixels, the calibrated pixel size was 2.66 Å on the specimen level. The total electron dose applied to a tilt series was 100 e$^-$ Å$^{-2}$. For each tilt series, images were acquired using a bidirectional scheme between −48° and 48° relative to the lamella starting from either 0° or 21°, with an incremental step of 3°. The defocus target was set to be -2–5 μm. At each tilt, the image was recorded as movies divided into eight subframes.

All movie frames were corrected with a gain reference collected in the same EM session. Movement between frames was corrected using MotionCor2 without dose weighting[53]. Initial reconstructions of all tilt series were performed using AreTomo[54], and these 3D tomograms were examined to exclude tomograms with bad target tracking, crystal ice, aberrant defocus, incomplete axonemes (less than five doublet microtubules caused by FIB–SEM milling) and unfavorable orientations (perpendicular to the tilt axis). Alignment of selected tilt series and tomographic reconstructions were then performed using Etomo[55]. The aligned tilt series were then CTF corrected using TOMOCTF[56], and the tomograms were generated using TOMO3D[57] (bin2; pixel size, 7.06 Å).

In total, we started with 24 milling grids of mouse sperm and obtained 200 lamellae. Tilt series with no crystal ice were kept and processed further. In some cases, parts of the (9 + 2) axoneme was milled away and we only processed the ones with at least five doublets and also enough space to include the full central pair complex for subvolume averaging. In the end, the final reconstructions of the consensus averages were from 69 usable tomograms. The per-doublet averages were based on subsets of the 69 tomograms (58–64 tomograms). For the human sperm cryo-ET dataset, the final reconstructions of the consensus averages are from 65 tilt series with the flagellar orientations within ~30° of the tilt axis. Note that the final tomograms used for different reconstructions vary as tomograms may be discarded due to poor alignment or may lack certain doublets.

### Subvolume averaging

Subsequent subvolume extraction, classification and refinement were all performed using RELION3 (ref. [58]) (schematic workflow in Extended Data Fig. 2a). Briefly, subvolumes from the doublets were manually picked every 24 nm and extracted with a box size large enough to accommodate a complete 96-nm-repeating unit (pixel size, 7.06 Å; box size, 180 pixels; dimension, 127.08 nm). Initially, subvolumes were aligned to a map of *Tetrahymena* doublets (EMD-9023) low-pass filtered to 80 Å, and the resulting map was used as the reference for further processing. Manual curation of the data was performed to check the alignment accuracy and data quality for each tomogram. Supervised 3D classification on RSs gave rise to four class averages of the 96-nm-repeating units at four different registries, staggered from one another by 24 nm. The pixel view in IMOD was used to determine the coordinates corresponding to the base of RS2 in all four class averages, and all subvolumes were re-extracted and recentered to the same point. All subvolumes were combined and aligned to one reference, and duplicate subvolumes were removed based on minimum distance (<40 nm). The remaining subvolumes were aligned to yield the consensus average for all nine doublets. These subvolumes were later remapped and sorted to calculate per-doublet averages (the script is also available below). The MIPs inside microtubule doublets repeat every 48 nm. Subvolumes of the 96-nm-repeating units were recentered on MIP features that repeat every 48 nm and refined with a mask focusing on the microtubules only.

To generate an average for the central pair complex, subvolumes were picked and extracted from the central pair complex every 16 nm (pixel size, 7.06 Å; box size, 160 pixels; dimension, 112.96 nm) (Extended Data Fig. 2b). Refinement of all subvolumes to an average of the sea urchin central pair (EMD-9385) resulted in a central pair with only 16-nm-repeating features. The alignment parameters were modified to reset all translations (*x, y, z*) to zero and then used for a second round

of refinement with local search only. Focused classification was then performed on the microtubule-associated proteins on the C1 microtubule to separate the two populations of subvolumes (~50% each) corresponding to the central pair complex with 32-nm periodicity and an offset of 16 nm between them. The subvolumes were then recentered and extracted on the same protein features for the two class averages. All subvolumes were aligned to the same reference using local searches. Duplicate subvolumes were removed based on the minimum separating distance (<30 nm). The remaining subvolumes were refined to generate the consensus average for the central pair complex.

Subvolumes large enough to include all nine doublets were re-extracted, centering on the aligned subvolumes of the central pair complex (pixel size, 21.18 Å; box size, 160 pixels; dimension, 338.88 nm). These subvolumes were averaged without further alignment and used to re-extract subvolumes corresponding to individual doublets. Subvolumes corresponding to a particular doublet were aligned and remapped back to the tomograms in three dimensions. However, these subvolumes do not have the correct alignment for the 96-nm-repeating features, but their centers nevertheless trace the axis of that microtubule doublet accurately when they are remapped in three dimensions (using the script available below). The 96-nm periodic subvolumes curated above to generate the consensus average of 96-nm repeats were then sorted. The subvolumes corresponding to a particular doublet were then aligned and averaged.

For multibody refinement (schematic workflow in Extended Data Fig. 2c), the subvolumes corresponding to the specific RS-central pair interface were re-extracted based on the subvolumes of 96-nm units of individual doublet microtubules. The particles were recentered at the junction between the head and the stalk of RS2 to include enough features from both the doublets and the central pair complex. The RSs were then aligned with a mask. This mask and a mask covering the central pair complex were used for the multibody analysis implemented in RELION3 (ref. [36]). Briefly, separate refinement of the RSs and central pair protrusions provided two sets of alignment parameters: three translational shifts ($x$, $y$, $z$) and three Euler angles required to align one subvolume to each reference. Thus, 12 parameters can be used to remap the two references back to each raw subvolume, and the spatial relationship of the two rigid bodies in each subvolume could be determined by these 12 parameters. Principal-component analyses essentially reprojected the original 12-dimensional data in a new 12-dimensional space with 12 new orthogonal eigenvectors. These 12 eigenvectors could be mathematically determined so that they represent decreasing variations of the entire data along the individual eigenvectors. Our analyses suggest that the first eigenvector or axis is parallel to the axoneme axis and the variations along this axis account for 40–50% of the total variations. When all subvolumes were divided into ten groups based on their projection values on the first axis (10% lowest, 10–20% lowest, …, 90–100% highest). Each group was then represented by a snapshot depicting the averaging projection values of the group members, and ten of these snapshots were morphed to generate the animation (Supplementary Video 1).

The resolutions for maps were estimated based on FSC values of two independently refined half datasets (FSC = 0.143). Local-resolution maps for the consensus averages of doublets and the central pair complex of both human and mouse sperm were calculated by blocres in Bsoft (Supplementary Fig. 2). These local-resolution maps represent relative differences in resolution across the maps, but the absolute values may not be exact. IMOD was used to visualize the tomographic slices[55]. UCSF Chimera was used to manually segment the maps for various structure features, and these maps were colored individually to prepare the figures using UCSF ChimeraX[59–61].

### Estimation of accumulated offsets in axonemes

The bending curvature of mouse sperm could be as large as $2 \times 10^5$ m$^{-1}$ or 0.20 µm$^{-1}$ based on literature (OD = 5 µm = 5,000 nm)[38]. The distance between the neighboring doublets is 72 nm (measured in the average of the axoneme shown in Fig. 2d, OB = 5,072 nm). If AB and CD represent 96-nm repeats from doublet 1 and doublet 2, respectively, offset $\Delta$ = CE ≈ OC × (∠COD − ∠AOB) = 5,000 × (96 ÷ 5,000 − 96 ÷ 5,072) = 1.4 nm.

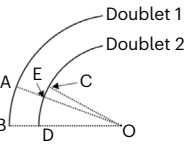

Note that the $\Delta$ represents the additional offset per 96-nm repeat. In a tomogram that contains an axoneme of ~2 µm, if the offset between the first pair of 96-nm repeats is 0 nm, the offset of the 20th pair is 28 nm. Note that this initial offset (0 nm) is tomogram specific depending on how much sliding has happened upstream or downstream of the imaging area and the imaged cell.

### Reporting summary

Further information on research design is available in the Nature Portfolio Reporting Summary linked to this article.

### Data availability

The maps of the following structures are available in the Electron Microscopy Data Bank: EMD-27444, consensus average of 96-nm-repeating structure of mouse doublets; EMD-27445, 32-nm-repeating structure of the central pair complex of mouse sperm; EMD-27446, mouse doublet 1; EMD-27447, mouse doublet 2; EMD-27448, mouse doublet 3; EMD-27449, mouse doublet 4; EMD-27450, mouse doublet 5; EMD-27451, mouse doublet 6; EMD-27452, mouse doublet 7; EMD-27453, mouse doublet 8; EMD-27454, mouse doublet 9; EMD-27455, 48-nm-repeating structure of the doublet microtubule of mouse sperm; EMD-27456, 5–6 bridge of mouse sperm; EMD-27462, consensus average of the 96-nm-repeating structure of human doublets; EMD-27463, 32-nm-repeating structure of the central pair complex of human sperm; EMD-27464, human doublet 1; EMD-27465, human doublet 2; EMD-27466, human doublet 3; EMD-27467, human doublet 4; EMD-27468, human doublet 5; EMD-27469, human doublet 6; EMD-27470, human doublet 7; EMD-27471, human doublet 8; EMD-27473, human doublet 9. The raw tilt series of mouse sperm lamellae and the corresponding tilt angle files are available in the EMPIAR database (EMPIAR-11221).

### Code availability

The script to remap coordinates of subvolumes in three dimensions is provided as Supplementary Code.

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

## Acknowledgements

We are grateful to members of the Vale and Agard laboratories for discussions and critical reading of the manuscript. We thank Shixin Yang from the cryo-EM facility at Janelia Research Campus for his assistance with data collection. We thank Caiying Guo for generously sharing the resources required for the mouse experiments. We thank Zanlin Yu, Eric Tse and David Bulkley in the University of California, San Fransisco (UCSF) EM core facility for their assistance with data collection. We thank Sam Li and Shawn Zheng at UCSF for suggestions on EM data processing. We thank Tom Goddard at UCSF for providing a script to mark coordinates of subtomograms. We used and appreciated computing resources at both the workstations at Janelia Research Campus and the HPC Facility at UCSF. Z.C. was supported by the Helen Hay Whitney Foundation Postdoctoral Fellowship. W.M.S. was supported by the National Science Foundation Graduate Research Fellowship Program under grant numbers DGE 1752814 and DGE 2146752. Any opinions, findings and conclusions or recommendations expressed in this material are those of the author(s) and do not necessarily reflect the views of the National Science Foundation. P.V.L. received funding from a Pew Biomedical Scholars Award and a GCRLE grant from the Global Consortium for Reproductive Longevity and Equality made possible by the Bia-Echo Foundation. D.A.A. received funding from NIH R35GM118099. R.D.V. received funding from NIH R35GM118106 and the Howard Hughes Medical Institute. The UCSF cryo-EM facility has been supported by NIH grants 1S10OD026881, 1S10OD020054 and 1S10OD021741.

## Author contributions

Z.C., G.A.G., D.A.A. and R.D.V. conceived the project and designed the experiments after discussions with other authors. S.Z. provided the mouse sperm sample. M.S. and Z.C. prepared mouse sperm lamellar grids. W.M.S. and P.V.L. provided human sperm samples. Z.C. and Y.L. prepared the human sperm grids. M.S., Z.C. and G.A.G. performed FIB–SEM processing. Z.C. and Z.Y. optimized data collection. Z.C. processed the data with help from M.S., X.Z. and R.Y. and suggestions from G.A.G. and D.A.A. Z.C. and R.D.V. wrote the manuscript draft with the help of comments from all authors.

## Competing interests

The authors declare no competing interests.

## Additional information

**Extended data** is available for this paper at https://doi.org/10.1038/s41594-022-00861-0.

**Correspondence and requests for materials** should be addressed to David A. Agard or Ronald D. Vale.

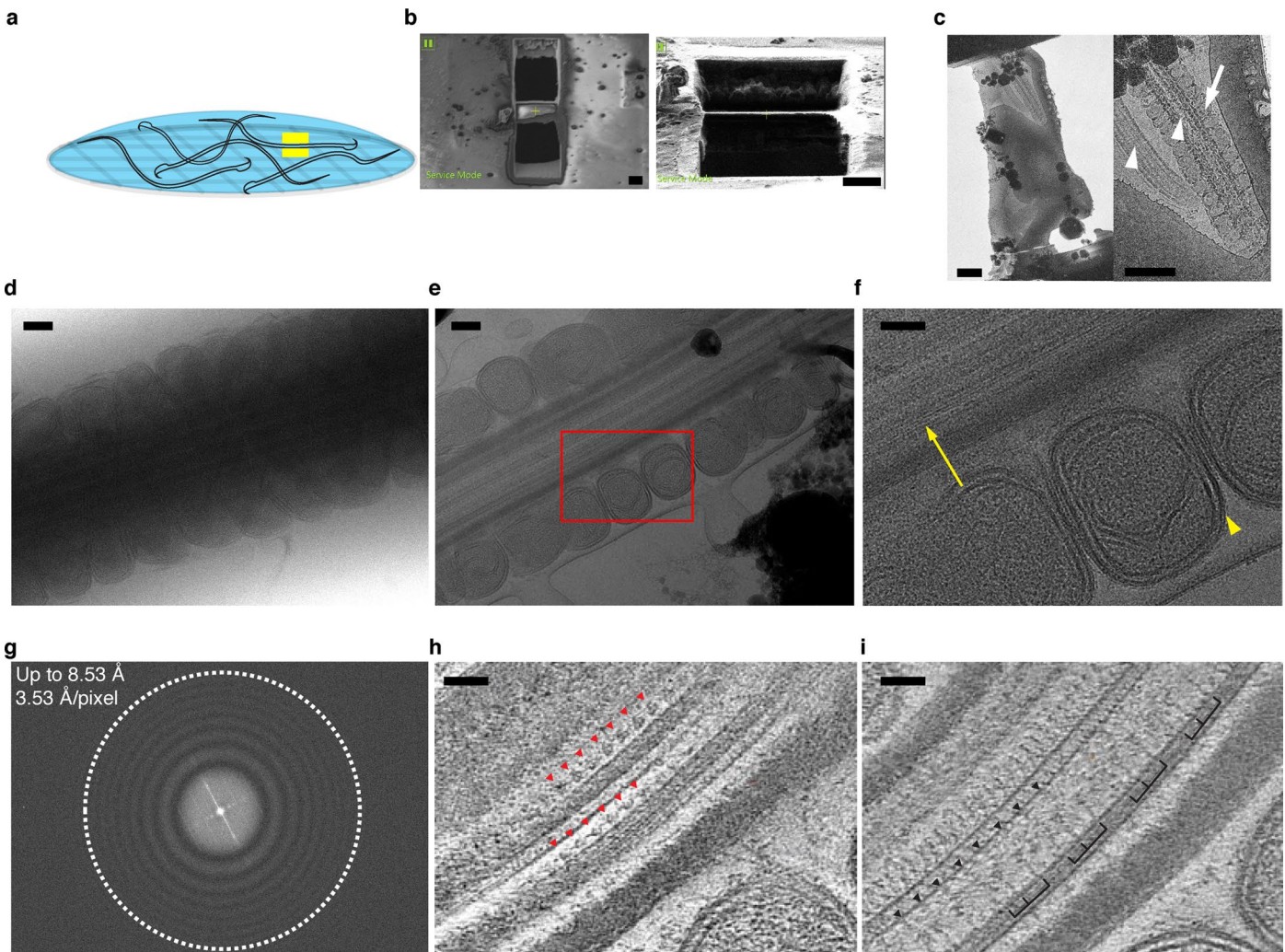

**Extended Data Fig. 1 | The workflow of FIB-SEM/cryo-ET and representative images. a**, Schematic of EM grids with vitrified mouse sperm. An example for where the windows were milled by FIB-SEM is denoted by two yellow boxes. **b**, SEM (left) and FIB (right) images of a lamella are shown (scale bar: 5 μm, N = 200 lamellae). **c**, A low-magnification montage and its zoom-in view of a lamella acquired using TEM (scale bar: 1 μm, N = 200 lamellae). Axonemes and mitochondria are indicated by white arrowheads and an arrow, respectively. **d**, A representative image of non-milled sperm axoneme recorded by 300 kV Titan Krios TEM at a total dose of 3 e⁻/Å² (scale bar: 100 nm, N = 10 tilt series, not processed further). **e**, A representative image of milled lamella (thickness ~300 nm) is shown (scale bar: 50 nm, N = 69 tilt series). The zoom-in region in **f**

is outlined (the red rectangle). **f**, Individual protofilaments of microtubules and double–bilayer membranes of the mitochondria are indicated by yellow arrows and arrowheads. **g**, Fourier transform of the image in **e** is shown and the upper limit of ctf fitting when using CTFFIND4 is indicated (at the pixel size of 3.53 Å, N = 69 zero-tilt images). **h, i**, Two sections of 3D tomograms are shown. Outer arm dyneins (every 24 nm) anchored on doublet microtubules are indicated in **h** (red arrowheads). Protrusions (every 32 nm) extending from the singlet microtubule are indicated in **i** (black arrowheads). Periodic patterns of three radial spokes on the doublet microtubules (every 96 nm) are grouped and indicated in **i** (black brackets) (scale bar: 50 nm, N = 69 tomograms).

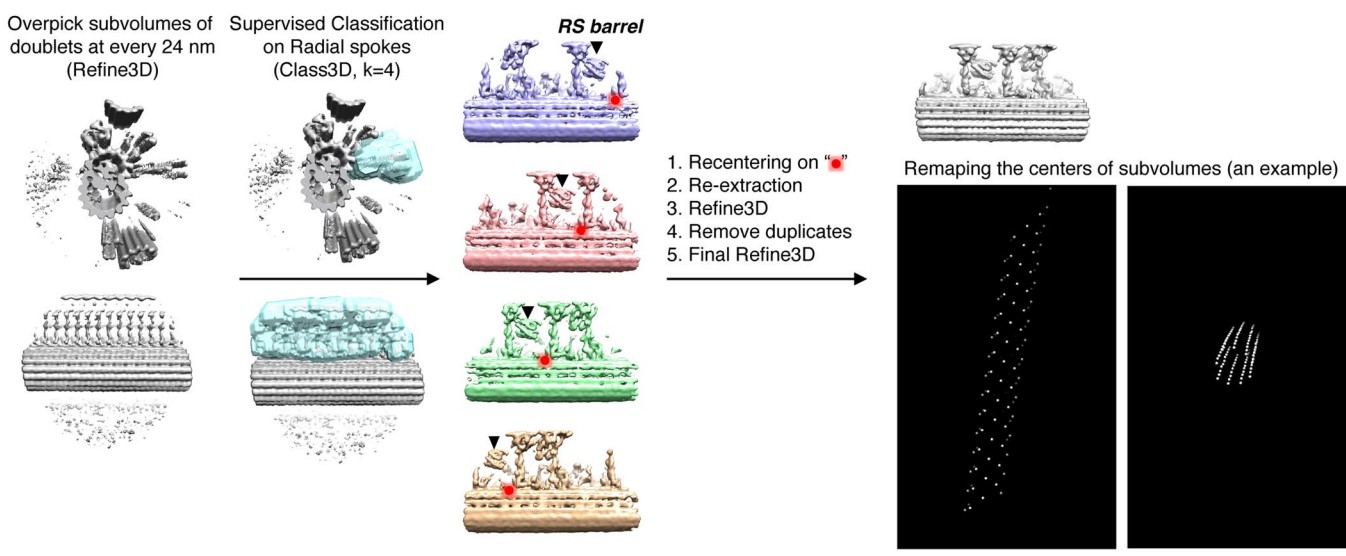

### a Processing workflow of 96-nm repeating units of doublets

Overpick subvolumes of doublets at every 24 nm (Refine3D)

Supervised Classification on Radial spokes (Class3D, k=4)

*RS barrel*

1. Recentering on "●"
2. Re-extraction
3. Refine3D
4. Remove duplicates
5. Final Refine3D

Remaping the centers of subvolumes (an example)

### b Processing workflow of 32-nm repeating units of central pair complex

Overpick subvolumes of central pair at every 16 nm (Refine3D)

Unsupervised Classification on C1 maps (Class3D, k=2)

1. Recentering on "●"
2. Re-extraction
3. Refine3D
4. Remove duplicates
5. Final Refine3D

Expansion of subvolumes (by three folds)

Doublet 1

### c Workflow of multibody analyses

1. Recentering on "●"
2. Re-extraction
3. Refine3D on RS

mask for CP

mask for doublets

Multibody analysis (RELION3)

Remaping the centers of subvolumes (an example)

Doublet 1

Doublet 1

**Extended Data Fig. 2 | Schematics of data processing. a**, A schematic of the processing workflow of 96 nm-repeating units of doublets to achieve the consensus average combining data from all nine doublets. An example of remapping of the centers of the final subvolumes in three dimensions is shown. **b**, A schematic of the processing workflow of 32 nm-repeating units of central pair complex to achieve the subvolume average. Recentering of subvolumes of the central pair complex (every 32 nm) on the doublet 1 is shown. Remapping of these subvolumes helps to trace the trajectory of doublet 1 in three dimensions. The centers of these doublet 1 subvolumes are shown in the same tomogram as in **a**. **c**, A schematic of the workflow of multi-body refinement.

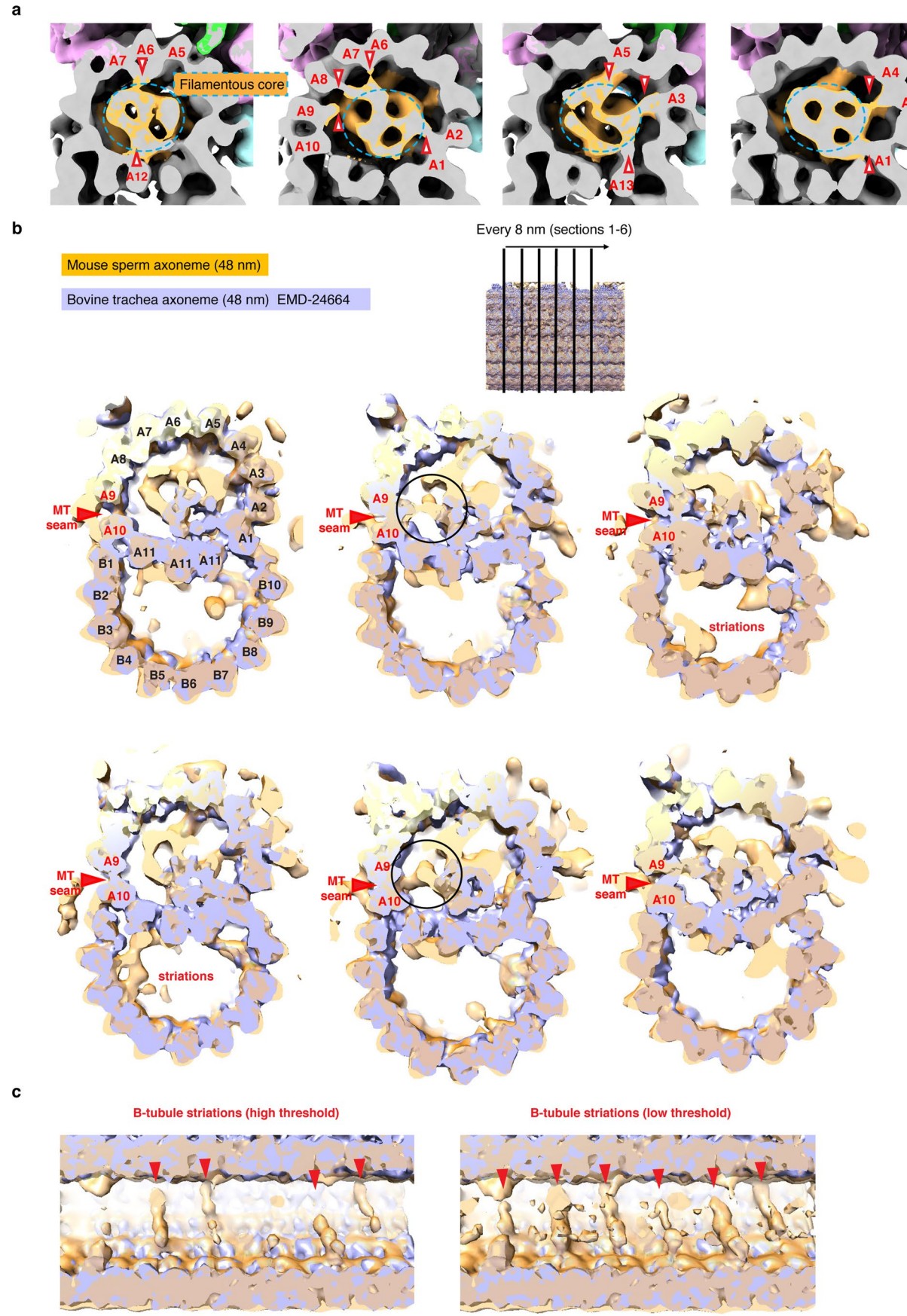

**Extended Data Fig. 3 | See next page for caption.**

**Extended Data Fig. 3 | MIPs network in A-tubule of doublet microtubules.**
**a**, Four cross-section views of the A tubule in the doublet microtubule are shown. The MIPs network is colored in orange. Three modes of interactions between the MIPs and the protofilaments are observed (see text). The protrusions from the filamentous core are indicated by empty red arrowheads and dashed cyan ovals, respectively. **b**, Six cross-section views of the overlay of the average of the 48-nm repeat of doublets of mouse sperm (orange, this study) and the 48-nm repeat of doublets of bovine trachea cilia (blue, EMD-24664)[26]. The contour levels of the two maps are adjusted so that the densities of microtubules are matched. The microtubule seam, A9 and A10 protofilaments are labeled in all views. Note there are additional densities near A9-A10 region in the mouse sperm axoneme (black circles). We observed density striations in the B-tubule in some cross sections. **c**, An orthogonal view of the B-tubule striations along the helical pitch of the microtubule at a high and a low threshold, respectively. Note the striations are separated by 8 nm and they cover the intradimeric interface between the α- and β-tubulins based on the model for the bovine doublet (PDB: 7rro)[26].

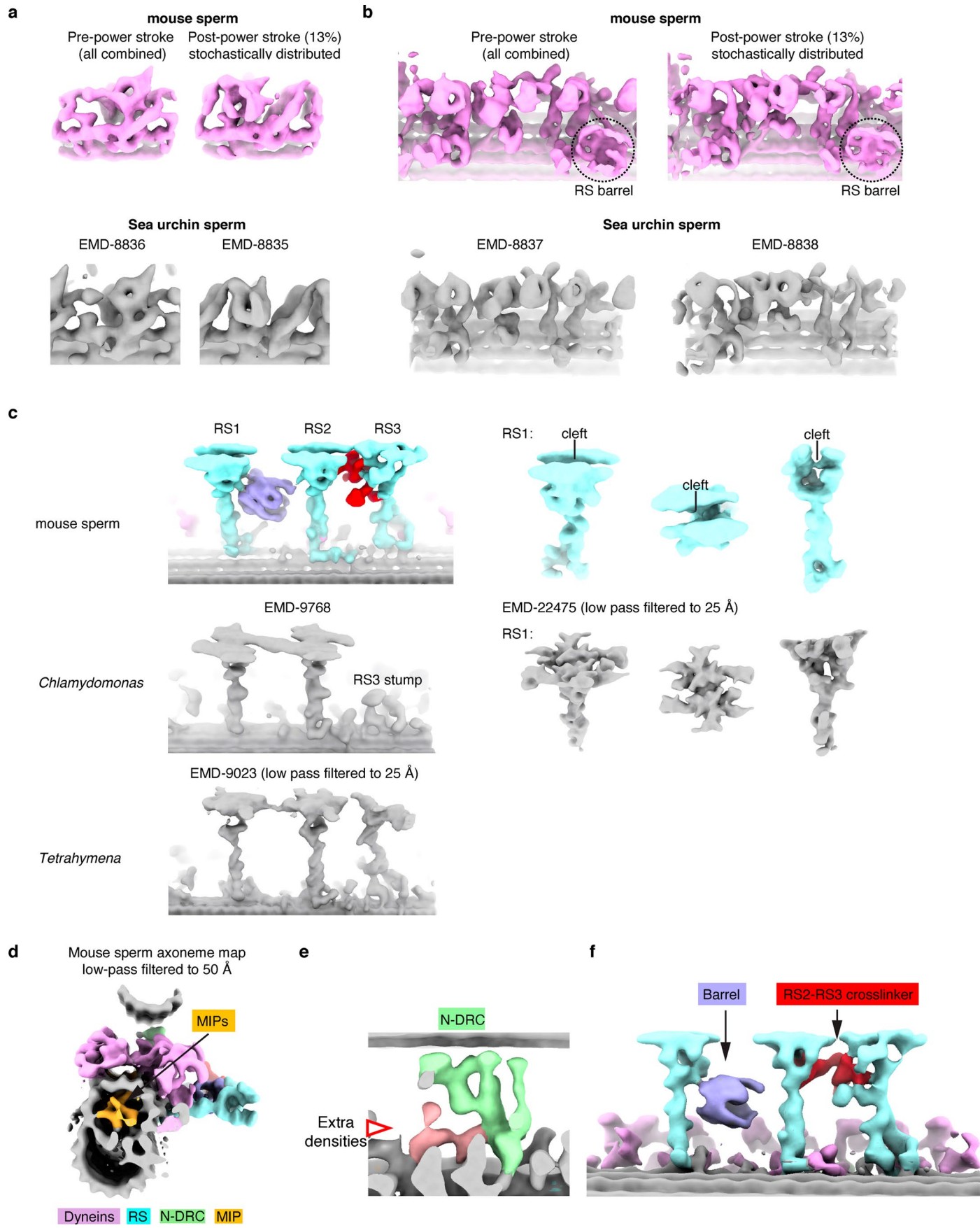

**Extended Data Fig. 4 | See next page for caption.**

**Extended Data Fig. 4 | Comparisons of dyneins and radial spokes of mouse sperm axonemes to published equivalent structures from other motile cilia. a**, Comparison of outer arm dyneins in mouse sperm axoneme (this study) and reported maps for those from sea urchin sperm (EMD-8835 and EMD-8836)[46]. **b**, Comparison of inner arm dyneins in mouse sperm axoneme (this study) and published maps for those from sea urchin sperm (EMD-8837 and EMD-8838)[46]. Note the barrel densities between RS1 and RS2 only exist in the mouse sperm are indicated (dashed ovals). **c**, Comparison of radial spokes of mouse sperm axoneme (this study) and the ones from motile cilia of *Chlamydomonas* (EMD-9768[30] and EMD-22475[14]) and *Tetrahymena*

(EMD-9023)[29]. Note the RS3 stump in *Chlamydomonas* is much shorter than the others and the head of RS3 from *Tetrahymena* has a smaller surface area. **d-f**, The doublet average of mouse sperm axoneme is low-pass filtered to 50 Å and viewed from different angles. Note this resolution is lower than published axoneme maps (for example 34 Å for EMD-5950 shown in Fig. 1 for comparison)[31]. The extensive network of MIPs (**d**), extra densities at N-DRC (**e**) and radial spokes (**f**) remain apparent while comparing to human respiratory cilia, suggesting the presence of novel densities in our maps are not due to higher resolutions, but most likely the presence of additional proteins.

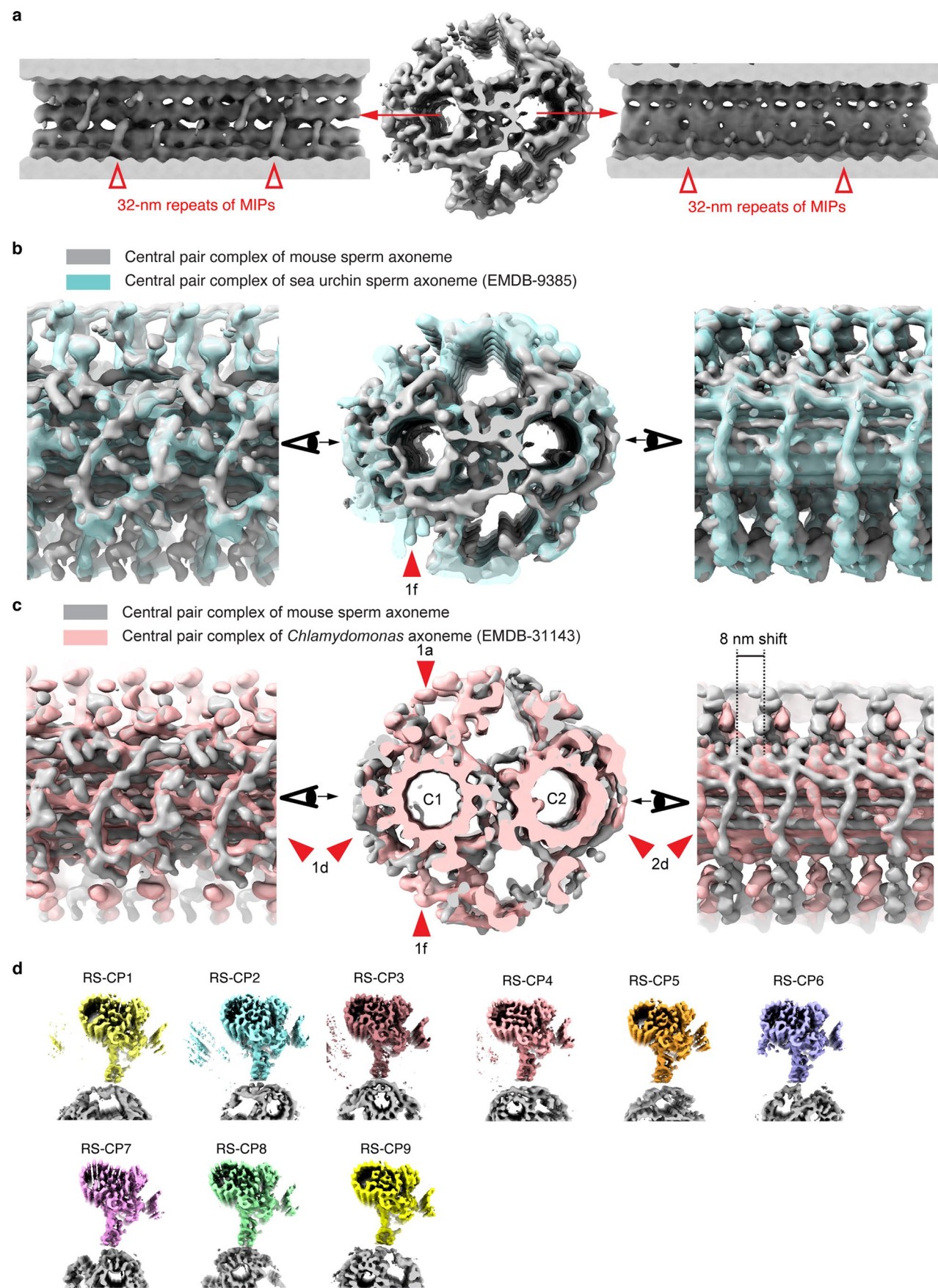

**Extended Data Fig. 5 | See next page for caption.**

**Extended Data Fig. 5 | The central pair complex in mouse axoneme possesses more MIPs but the overall shape is similar to the one from sea urchin sperm.**
**a**, Two longitudinal views of the MIPs inside the two singlet microtubules are shown. Periodic features (every 32 nm) are indicated (red empty arrowheads). **b**, The average of the central pair complex in mouse sperm axonemes is overlaid on the map for the central pair complex from sea urchin sperm (EMD-9385) and they are colored in grey and cyan, respectively[25]. Note most of the protrusions are very similar and the sea urchin-specific protrusion 1f is indicated in the middle panel (red arrowhead). **c**, The average of the central pair complex in mouse sperm axonemes is overlaid on the map for the central pair complex from *Chlamydomonas* (EMD-31143)[35]. Note the maps are aligned based on the similar C1 protrusions 1d. Other protrusions are generally less similar (1a, 1f and 2d) and there appears to be an 8-nm shift for the C2 protrusions. The *Chlamydomonas* C2 microtubule was recently shown to possess two possible registers with 8-nm longitudinal differences[33]. **d**, The spatial relationship between the nine radial spokes and the central pair complex are determined by multibody refinement and the 'average positioning' of these two rigid bodies are shown (RS-CP1 denotes the interface between radial spokes from doublet 1 and the central pair.

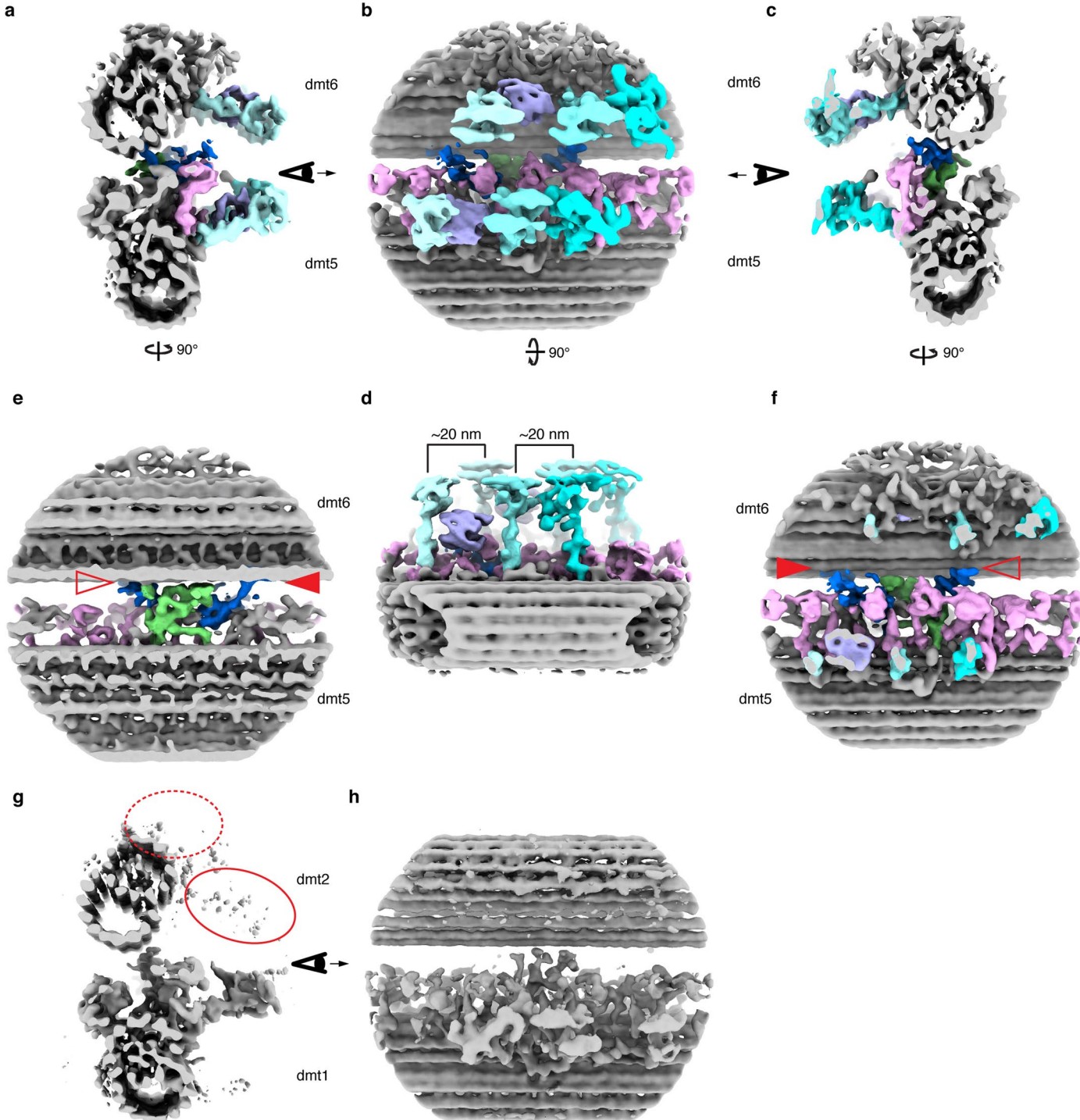

**Extended Data Fig. 6 | The 5-6 bridge of mouse sperm axonemes. a-d,** Four views of the composite map for 5-6 bridge in mouse sperm axonemes combining maps for local refinement of the doublet 5, doublet 6 and the bridge. The inner arm dyneins, radial spokes, the barrel, N-DRC and extra bridge densities are colored in pink, different shades of cyan, light blue, green and blue, respectively. Note that the radial spokes which repeat every 96 nm on doublet 5 and doublet 6 are resolved and there appears to be a ~20 nm offset between their longitudinal registers, as indicated in the staggering distance of RS1 and RS2 from doublet 5 and doublet 6 in **d. e, f,** Two cut-in views of **a** and **c** are shown. Bridge−specific densities are highlighted in blue. These densities extend from the inner arm dyneins of doublet 5 towards the microtubule of doublet 6. Direct contacts of the bridge-specific densities to doublet 6 is indicated by the solid red arrowhead while densities in close proximity of the base of radial spoke 2 in doublet 6 are indicated by the empty red arrowhead. **g, h,** Subvolumes for doublet 1 was aligned and equivalent views of doublet 1 and 2 are shown (as in panel **a** and **b**). The expected positions for dyneins and radial spokes on doublet 2 are circled by dashed and solid red ovals. These 96 nm-repeating structures on doublet 2 are not resolved even after local refinement, suggesting the offsets between doublet 1 and doublet 2 are not consistent. The microtubule from doublet 2 is nevertheless partially resolved since variant longitudinal mismatching would not smear a tube of 'continuous densities' compared to 'discrete densities' like dyneins and radial spokes through averaging.

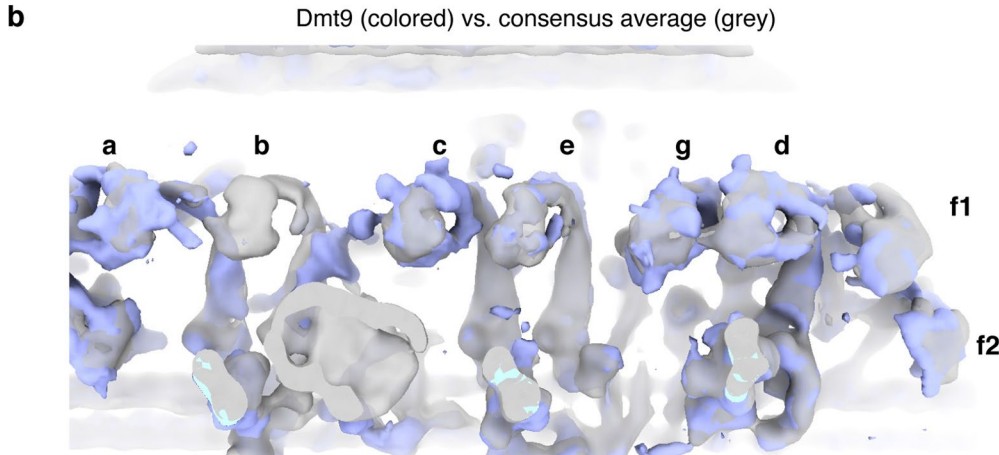

**a** Dmt5 (colored) vs. consensus average (grey)

**b** Dmt9 (colored) vs. consensus average (grey)

**Extended Data Fig. 7 | The inner arm dyneins in doublet 5 and doublet 9 differ from the consensus average in mouse sperm axonemes. a**, An overlay of inner arm dyneins from doublet 5 and the concensus average. The assigned inner arm dyneins, N-DRC and 5–6 bridge densities from the doublet 5 are colored in pink, green and blue, respectively while the densities for the consensus average are in grey. Different inner arm dyneins are named according to the nomenclature defined by studies of *Chlamydomonas* flagella. The shifted positions of dynein e and dynein g are indicated by the two yellow arrows. **b**, An overlay of inner arm dyneins from doublet 9 and the consensus average were blue and grey, respectively. Note dynein b in doublet 9 is not resolved.

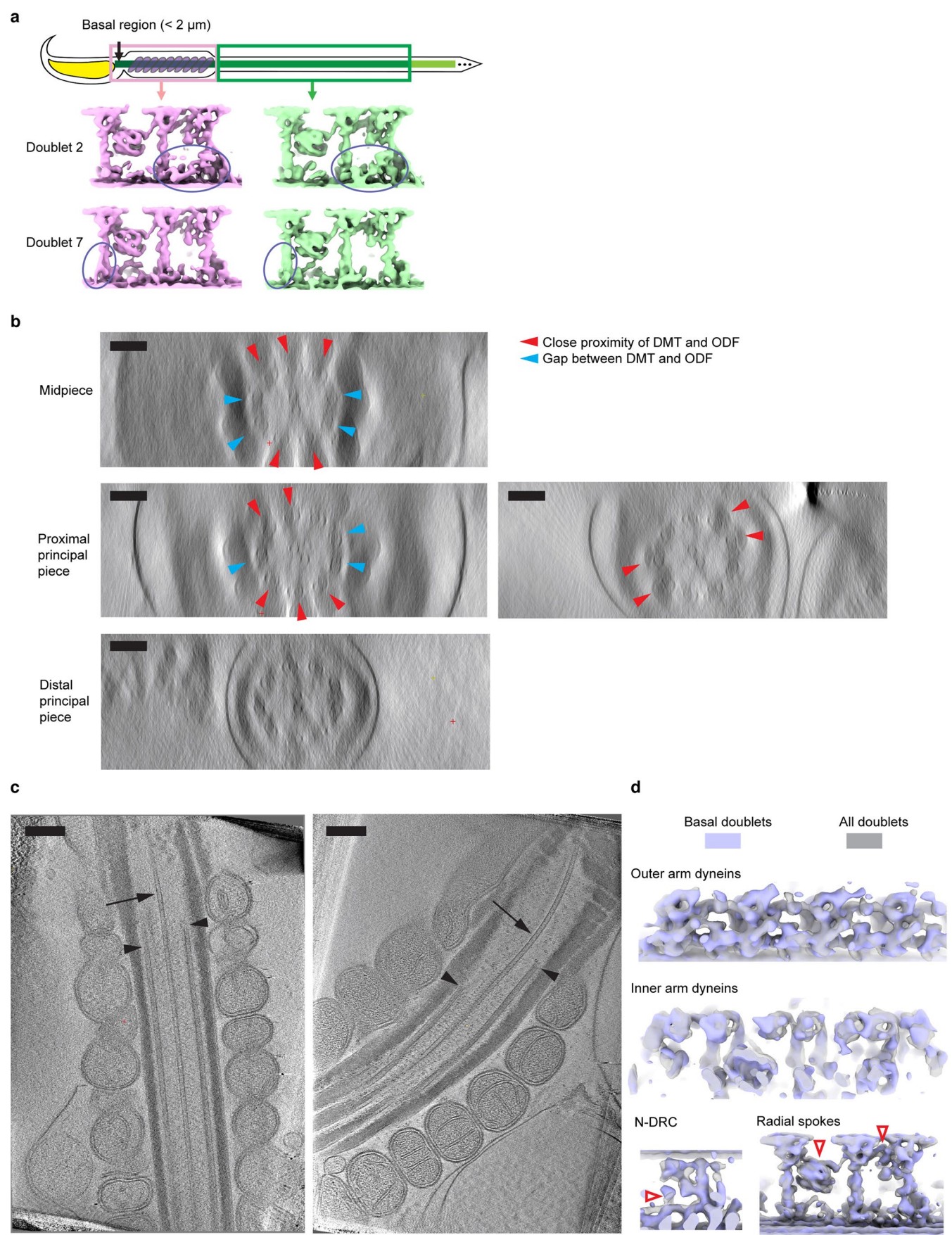

**Extended Data Fig. 8 | See next page for caption.**

**Extended Data Fig. 8 | Longitudinal consistency of doublets in mouse sperm axonemes. a**, Schematic of a mouse sperm flagella with the basal region (< 2 µm), midpiece and principal piece highlighted in the top panel. Subvolume averages of doublet 2 and doublet 7 from the midpiece and the principal piece are shown. The shapes of densities for the scaffolds at the base of radial spoke 3 appear different in doublet 2 and are circled. Additional densities at the base of radial spoke 1 in the doublet 7 in the midpiece region are observed and circled. **b**, Four xz slices of tomograms in different regions of the mouse sperm flagella are shown. Outer dense fibers that are in close proximity to the doublets and the ones with gaps in between are indicated (red and blue arrowheads, respectively)

(scale bars: 100 nm). **c**, Two xy slices of two 3D tomograms of the basal region of mouse sperm axoneme are shown. The singlet microtubules of the central pair complex extend further into the cell bodies and the beginning of the doublet microtubules are indicated by the arrows and arrowheads, respectively (scale bars: 200 nm). **d**, Densities corresponding to different protein complexes from subvolume averages of the basal region (6 tomograms, N = 670 subtomograms) and the consensus average are overlaid and colored in blue and grey, respectively. Note the sperm-specific features in N-DRC and radial spokes exist in the average of doublets in the basal region and are highlighted using empty red arrowheads.

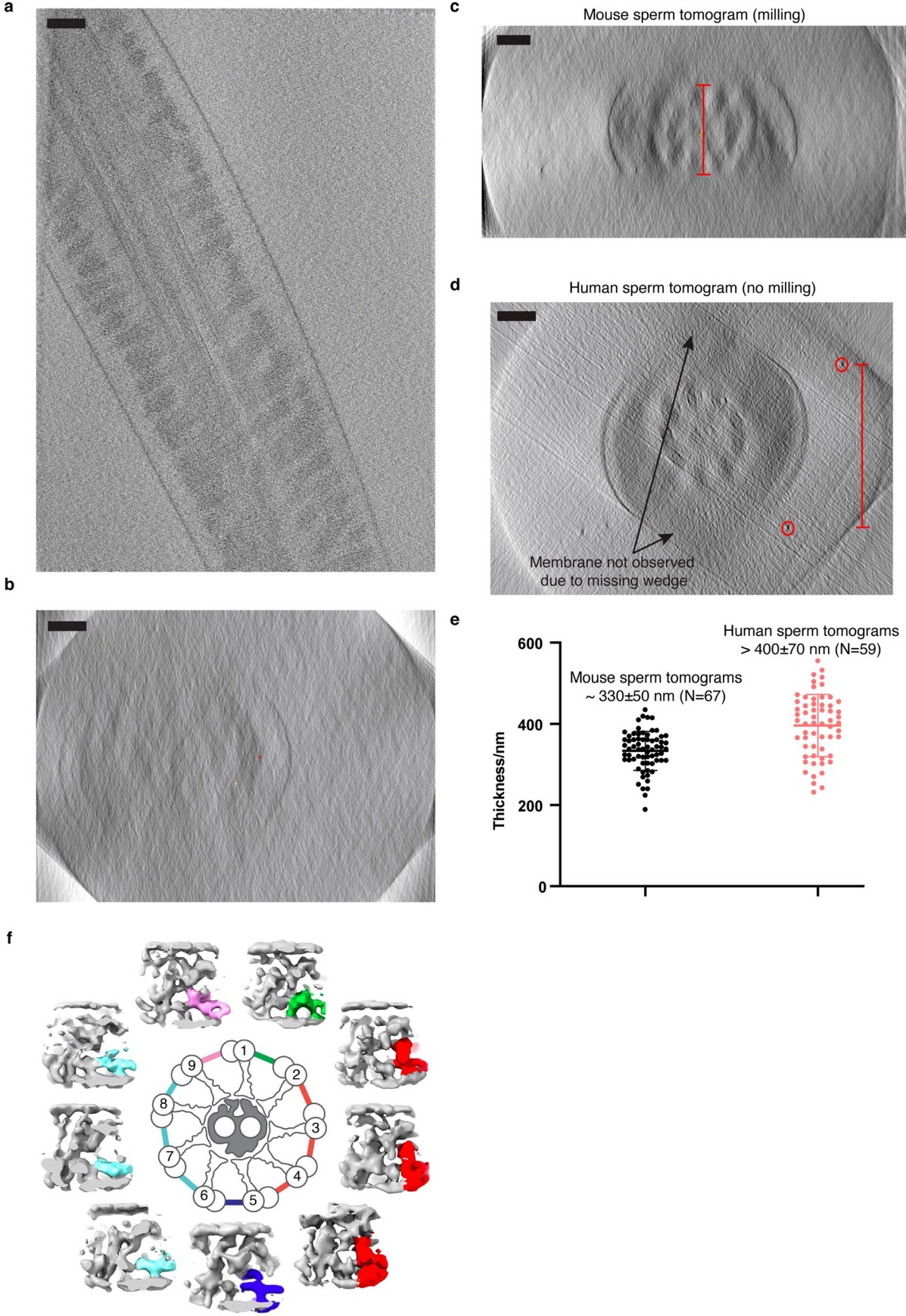

c Mouse sperm tomogram (milling)

d Human sperm tomogram (no milling)

Membrane not observed
due to missing wedge

e

Human sperm tomograms
> 400±70 nm (N=59)

Mouse sperm tomograms
~ 330±50 nm (N=67)

**Extended Data Fig. 9 | See next page for caption.**

**Extended Data Fig. 9 | CryoET data of human sperm without milling and asymmetric distribution of sperm-specific features in N-DRCs in human sperm. a**, A representative xy slice of a tomogram of human sperm (pixel size=10.648 Å, image thickness: 1, scale bar: 100 nm). Although the A- and B-tubule of the microtubule doublets are discernible, other features like axonemal dyneins and radial spokes are not clear in this slice (in contrast to Extended Data Fig. 1h-i). **b**, A representative xz slice of the same tomogram of human sperm (pixel size=10.648 Å, image thickness: 5, displayed in IMOD) (scale bar: 100 nm). **c**, A representative xz slice of a tomogram of mouse sperm lamella (pixel size=14.12 Å, image thickness: 10, displayed in IMOD). The red bar indicates how we measure the thickness of the lamella (scale bar: 100 nm). **d**, A representative xz slice of a tomogram of human sperm (pixel size=10.648 Å,

image thickness: 100, displayed in IMOD). Gold beads on the top and bottom surfaces are circled and the red bar indicates how we measure the thickness of the ice near the cell. The top and bottom membrane of the sperm cells is not resolved due to the missing wedge so the actual thickness is larger than the measured thickness of ice in the vicinity (scale bar: 100 nm). **e**, A dot plot showing the thickness of lamellae of mouse sperm and ice near human sperm as a lower bound estimate of the actual cells. The means and standard deviations are indicated (for N = 69 mouse sperm tomograms and N = 59 human sperm tomograms). **f**, Densities corresponding to N-DRCs from the nine per-doublet averages are shown around a schematic of the (9 + 2)- axoneme. Common features among the N-DRC are colored in grey, while the unique features are highlighted in various colors.

# Reporting Summary

## Statistics

For all statistical analyses, confirm that the following items are present in the figure legend, table legend, main text, or Methods section.

| n/a | Confirmed | |
|---|---|---|
| ☐ | ☒ | The exact sample size (*n*) for each experimental group/condition, given as a discrete number and unit of measurement |
| ☐ | ☒ | A statement on whether measurements were taken from distinct samples or whether the same sample was measured repeatedly |
| ☒ | ☐ | The statistical test(s) used AND whether they are one- or two-sided<br>*Only common tests should be described solely by name; describe more complex techniques in the Methods section.* |
| ☒ | ☐ | A description of all covariates tested |
| ☒ | ☐ | A description of any assumptions or corrections, such as tests of normality and adjustment for multiple comparisons |
| ☒ | ☐ | A full description of the statistical parameters including central tendency (e.g. means) or other basic estimates (e.g. regression coefficient) AND variation (e.g. standard deviation) or associated estimates of uncertainty (e.g. confidence intervals) |
| ☒ | ☐ | For null hypothesis testing, the test statistic (e.g. *F*, *t*, *r*) with confidence intervals, effect sizes, degrees of freedom and *P* value noted<br>*Give P values as exact values whenever suitable.* |
| ☒ | ☐ | For Bayesian analysis, information on the choice of priors and Markov chain Monte Carlo settings |
| ☒ | ☐ | For hierarchical and complex designs, identification of the appropriate level for tests and full reporting of outcomes |
| ☒ | ☐ | Estimates of effect sizes (e.g. Cohen's *d*, Pearson's *r*), indicating how they were calculated |

*Our web collection on statistics for biologists contains articles on many of the points above.*

## Software and code

Policy information about availability of computer code

| Data collection | SerialEM3.8.4 |
|---|---|
| Data analysis | The EM software used: RELION v1.4, v3.0.6, Motioncorr v2; Structural visualization: Chimera 8.6.1, ChimeraX 1.1, IMOD 4.9.13 |

For manuscripts utilizing custom algorithms or software that are central to the research but not yet described in published literature, software must be made available to editors and reviewers. We strongly encourage code deposition in a community repository (e.g. GitHub). See the Nature Portfolio guidelines for submitting code & software for further information.

## Data

Policy information about availability of data

All manuscripts must include a data availability statement. This statement should provide the following information, where applicable:

- Accession codes, unique identifiers, or web links for publicly available datasets
- A description of any restrictions on data availability
- For clinical datasets or third party data, please ensure that the statement adheres to our policy

The maps of the following structures are available in the Electron Microscopy Data Bank: EMD-27444, consensus average of 96 nm-repeating structure of mouse doublets; EMD-27445, 32 nm-repeating structure of central pair complex of mouse sperm; EMD-27446, mouse doublet 1; EMD-27447, mouse doublet 2; EMD-27448, mouse doublet 3; EMD-27449, mouse doublet 4; EMD-27450, mouse doublet 5; EMD-27451, mouse doublet 6; EMD-27452, mouse doublet 7; EMD-27453, mouse doublet 8; EMD-27454, mouse doublet 9; EMD-27455, 48 nm-repeating structure of doublet microtubule of mouse sperm; EMD-27456, 5-6 bridge of mouse sperm; EMD-27462, consensus average of 96 nm-repeating structure of human doublets; EMD-27463, 32 nm-repeating structure of central pair complex of human sperm; EMD-27464, human doublet 1; EMD-27465, human doublet 2; EMD-27466, human doublet 3; EMD-27467, human doublet 4;

EMD-27468, human doublet 5; EMD-27469, human doublet 6; EMD-27470, human doublet 7; EMD-27471, human doublet 8; EMD-27473, human doublet 9. The 69 raw tilt series of mouse sperm lamellae and the corresponding tilt angle files are available in the EMPIAR database (EMPIAR-11221).

# Field-specific reporting

Please select the one below that is the best fit for your research. If you are not sure, read the appropriate sections before making your selection.

☒ Life sciences ☐ Behavioural & social sciences ☐ Ecological, evolutionary & environmental sciences

For a reference copy of the document with all sections, see nature.com/documents/nr-reporting-summary-flat.pdf

# Life sciences study design

All studies must disclose on these points even when the disclosure is negative.

| | |
|---|---|
| Sample size | The mouse sperm cryoEM datasets were collected in 6 independently prepared samples and no statistical methods were used to predetermined sample size. The data were processed in batches and the biological structures were observed to be highly consistent between batches. Each EM grids usually contains millions of sperm cells and we randomly sampled cells for cryoET imaging. In total, 69 sperm/regions were stochastically sampled and all subtomogram averages were performed with N>700 particles from > 50 axoneme segments. We determined this to be sufficient owing to the consistency of the averages from three batches of datasets. Human sperm from a volunteer were collected and the cells were were normozoospermic with typical motility and morphologies. |
| Data exclusions | Through 3D classification procedures, we discarded "duplicated particles" that result from initial overpicking of particles based on their positions in the raw data. The procedure ensures that we only counted one subtomogram once in our data and this is standard and established in the cryoEM field. |
| Replication | The data were independently processed in 3 batches and similar results were obtained (with varied levels of noise depending on the size of the batch). All data were combined and used to make the figures. |
| Randomization | For cryoET imaging, there are millions of sperm cells on each grid and individual sperm were selected at random without bias toward sample location, size or fields under the microscope. For the data processing, we selected particles corresponding to specific cellular structures, which is standard in the cryoEM field. The gold-standard Fourier shell correlation was calculated from two independently refined half-maps, calculated based on two randomly selected halves of data. |
| Blinding | At the begining of study, we assume that we do not know the true mammalian sperm structures. Initial reference of the equivalent structures from Tetrahymena cilia and sea urchin sperm were low-pass filtered to 60 angstrom to eliminate the reference bias. Still, sperm-specific densities were found in the final reconstruction at ~25 angstrom resolution (higher than the reference), suggesting that these signals came from the actual data but not the initial reference. |

# Reporting for specific materials, systems and methods

We require information from authors about some types of materials, experimental systems and methods used in many studies. Here, indicate whether each material, system or method listed is relevant to your study. If you are not sure if a list item applies to your research, read the appropriate section before selecting a response.

### Materials & experimental systems

| n/a | Involved in the study |
|---|---|
| ☒ | ☐ Antibodies |
| ☒ | ☐ Eukaryotic cell lines |
| ☒ | ☐ Palaeontology and archaeology |
| ☐ | ☒ Animals and other organisms |
| ☐ | ☒ Human research participants |
| ☒ | ☐ Clinical data |
| ☒ | ☐ Dual use research of concern |

### Methods

| n/a | Involved in the study |
|---|---|
| ☒ | ☐ ChIP-seq |
| ☒ | ☐ Flow cytometry |
| ☒ | ☐ MRI-based neuroimaging |

## Animals and other organisms

Policy information about studies involving animals; ARRIVE guidelines recommended for reporting animal research

| | |
|---|---|
| Laboratory animals | C57Bl/6J mice were housed on a free-standing, individually ventilated (~60 air changes hourly) rack (Allentown Inc, Allentown, NJ). The holding room was ventilated with 100% outside filtered air with at 15 to 20 air changes hourly. Each ventilated cage (Allentown) was provided with corncob bedding (Shepard Specialty Papers, Milford, NJ), at least 8g of nesting material (Bed-r'Nest, The Andersons, Maumee, OH), and red Mouse Tunnel (Bio-Serv, Flemington, NJ). Mice were maintained on a 12:12-h light:dark cycle. The holding room temperature was maintained at 21±1°C with a relative humidity of 30% to 70%. Irradiated rodent laboratory chow (LabDiet 5053) was provided ad libitum and chlorinated (1-3ppm) RO water was provided without restriction. |

| Wild animals | No wild animals were used in the study. |
|---|---|
| Field-collected samples | No field collected samples were used in the study. |
| Ethics oversight | All mice were cared for in compliance with the Guide for the Care and Use of Laboratory Animals. All experiments were approved by the Janelia Research Campus' (JRC) IACUC. JRC is an AAALAC-accredited institution. |

Note that full information on the approval of the study protocol must also be provided in the manuscript.

# Human research participants

Policy information about studies involving human research participants

| Population characteristics | A man aged 25-39 years old were recruited and consented to participate in this study. Freshly ejaculated semen samples were obtained by masturbation. All processed samples were normozoospermic with a cell count of at least 30 million sperm cells per milliliter. |
|---|---|
| Recruitment | Healthy volunteers were recruited randomly and provided informed consent. All volunteers showed normal semen analysis at the time of sample collection. |
| Ethics oversight | The experimental procedures utilizing human-derived sperm samples were approved by the Committee on Human Research at the University of California, Berkeley, IRB protocol number 2013-06-5395. |

Note that full information on the approval of the study protocol must also be provided in the manuscript.

