## [Peer Review File · Nature Structural & Molecular Biology]

Peer Review Information

Journal: Nature Structural and Molecular Biology

Manuscript Title: In situ cryo-electron tomography reveals the asymmetric architecture of mammalian sperm axonemes

Corresponding author name(s): Dr Ronald Vale

Editorial Notes:

Reviewer Comments & Decisions:

Decision Letter, initial version:
--

Dear Dr. Vale,

Please accept my apologies for the unusual delay in reaching a decision on your manuscript "In situ cryo-electron tomography reveals the asymmetric architecture of mammalian sperm axonemes". I am afraid that with the recent changes to the editorial team, it took us much longer than usual to reach a decision. Nevertheless, we now have comments (below) from the 3 reviewers who evaluated your paper. In light of those reports, we remain interested in your study and would like to see your response to the comments of the referees, in the form of a revised manuscript.

You will see that the referees appreciate the results and find the conclusions timely and of wide interest. There are, however, several concerns and suggestions that should be addressed in a revision. While we agree with the referee #3 that identifying the new protein densities would be out of scope of a normal revision, this point should be addressed with additional discussion, in addition to addressing

the other technical comments of the reviewers.

Please be sure to address/respond to all concerns of the referees in full in a point-by-point response and highlight all changes in the revised manuscript text file. If you have comments that are intended for editors only, please include those in a separate cover letter. We are committed to providing a fair and constructive peer-review process. Do not hesitate to contact us if there are specific requests from the reviewers that you believe are technically impossible or unlikely to yield a meaningful outcome.

We expect to see your revised manuscript within 6 weeks. If you cannot send it within this time, please contact us to discuss an extension; we would still consider your revision, provided that no similar work has been accepted for publication at NSMB or published elsewhere.

Reporting Summary:

When submitting the revised version of your manuscript, please pay close attention to our [href="https://www.nature.com/nature-research/editorial-policies/image-integrity">Digital Image Integrity Guidelines. and to the following points below:](https://www.nature.com/nature-research/editorial-policies/image-integrity)

Please note that all key data shown in the main figures as cropped gels or blots should be presented in uncropped form, with molecular weight markers. These data can be aggregated into a single supplementary figure item. While these data can be displayed in a relatively informal style, they must

refer back to the relevant figures. These data should be submitted with the final revision, as source data, prior to acceptance, but you may want to start putting it together at this point.

Data availability: this journal strongly supports public availability of data. All data used in accepted papers should be available via a public data repository, or alternatively, as Supplementary Information. If data can only be shared on request, please explain why in your Data Availability Statement, and also in the correspondence with your editor. Please note that for some data types, deposition in a public repository is mandatory - more information on our data deposition policies and available repositories can be found below:

<https://www.nature.com/nature-research/editorial-policies/reporting-standards#availability-of-data>

[Redacted]

Note: This URL links to your confidential home page and associated information about manuscripts you may have submitted, or that you are reviewing for us. If you wish to forward

this email to co-authors, please delete the link to your homepage.

Sincerely,

Carolina

Carolina Perdigoto, PhD
Chief Editor
Nature Structural & Molecular Biology
orcid.org/0000-0002-5783-7106

Referee expertise:

Referee #1: cryo-ET/EM

Referee #2: cilia protein assembly

Referee #3: cryo-ET

Reviewers' Comments:

Reviewer #1:

Remarks to the Author:

Motile cilia, beating organelles powered by sliding of dynein ATPase on microtubules, provide various interesting topics, such as biophysical mechanism of beating motion, networking of >400 component proteins, mechanism to form the highly conserved (from green algae to human) 9+2 axoneme, and medical relevance (primary ciliary dyskinesia). For those questions, precise description of axonemal structure leading to 3D mapping of component proteins is essential.

This manuscript, Chen et al., presents detailed 3D cellular structure of the axoneme from mammalian sperm, employing cryo-FIB milling, cryo-electron tomography (cryo-ET). Mammalian sperm axoneme is covered by thick sheath, which prevented cryo-EM observation. Recently developed cryo-FIB milling enabled thin slicing of frozen sperm to visualize the axoneme. This work is the first cryo-ET work of mammalian sperm axoneme. With intermediate resolution (20~25Å) cryo-ET analysis and subtomogram averaging, the authors successfully visualized the 96nm periodic unit from the mammalian sperm axonemes and discovered a number of structures which were not found in other species or mammalian respiratory cilia, inside the microtubule, beside N-DRC and between radial spokes 1 and 2. Especially they are interested in difference of nine doublet microtubules. It was pointed out that the proteins binding on the nine microtubules are not the same. First this asymmetry was found in outer dynein arm missing in doublet 1 from *Chlamydomonas* (Hoops and Witman, 1983 JCB). Bui et al. (2008) JCB revealed more detail of this asymmetry using cryo-ET and proved that there are inter-doublet linkers localized in particular pair of doublets. The present work proved that many of sperm specific structures appear differently between nine doublets. Asymmetry exists not

only the protein structure on the doublet, but also the interface between the radial spoke and the central pair – only doublet 8 interacts with CP directly. Since the beating motion of sperm axoneme is asymmetric (planar), they attempted to relate the structural asymmetry and functional asymmetry. The work is novel with careful analysis properly using high-end technique (such as cryo-FIB and PCA of 3D subtomograms) and presenting beautiful figures. Undoubtedly it deserves publication in this level of journal, although this reviewer is still wondering if cell biology journals (Nat. Cell Biol., J. Cell Biol., etc) or more general journals (eLife, PNAS) may attract more readers for this manuscript, since this work describes global outline of protein complexes but does not address any particular identified proteins nor mention 3D conformation of molecules. While it is not known which proteins the features described here correspond to, this work will inspire molecular/cellular biologists to do protein identification. Here is comments, which, this reviewer believes, will help in case this manuscript will be published in Nat. Struct. Mol. Biol. or otherwise any journal.

Major points:

Line180-182, Fig.2d,e: This reviewer is not sure whether the averaging was done properly to derive the conclusion (doublets longitudinally not aligned each other). Since CP and doublets have 32nm and 96nm periodicity, if the alignment/averaging was done using CP, doublets, even if they are aligned, will be averaged out between the three (=96nm/32nm) possible shifts. To make sure if the doublets are aligned each other, they must be aligned as the whole axoneme (9+2) and averaged. Probably their conclusion is correct in the end, but from Fig.2 it cannot be concluded so.

Line238-241, Extended Data Fig.7: indexing of single-headed inner dyneins (after Chlamydomonas) is wrong. It should be from the proximal end dynein a, b, c, e, g, d (they indexed a, b, c, d, e, g). Please refer Bui et al. 2012 JCB.

Minor points:

Line60-61: Bridge like structures between two adjacent doublets, which do not follow 9-fold symmetry, were pointed out in Chlamydomonas (Bui et al. 2009 JCB).

Line114: "Protrusions extending from filamentous structure": Please indicate them in Extended Data Fig. 2a.

Line140-142: Unique structure of human RS heads: does this structure have correlation to the difference between the authors structure of reconstituted RS head and Brown's structure?

Line177: "for the center central pair complex": it is not clear.

Line209-213: This reviewer could not follow the description. Indications in Fig.3d will help.

Line394, Fig.7d: It will be helpful if it is shown how much displacement should happen in the absence of 5-6 linker with normal axoneme curvature during the beating.

Fig.7c: What are red arrows?

Line577: This should be for panel g.

Methods: How principal component analysis of subtomograms was done should be written.

Reviewer #2:

Remarks to the Author:

Major comments

Key results:

Identification of novel barrel shaped structures and connections between radial spoke proteins which are asymmetrically arranged leading to the proposed asymmetric beat pattern specific to mammalian sperm

Originality and significance:

The conclusions are original and the identification of asymmetrically positioned novel barrel-like structures as well as radial spoke connections among other observations is interesting.

Specific to the findings on per-doublet averaging and connections to the central pair, this reviewer would point the authors to the following preprint 'Cryo-EM structure of an active central apparatus' DOI: <https://doi.org/10.1101/2022.01.23.477438>. This preprint reports repeating surface densities on high resolution models of the central pair apparatus very similar to the one's highlighted in figure 2 and extended data figure 5 of the study under review. The authors of the current study could benefit from a more detailed comparison with the structural models presented in the suggested preprint. Such a comparison could reveal unique features specific to mammalian sperm central pairs.

Data & methodology, statistics, conclusions:

The study finds new structures which are of interest to structural biologists working on broad aspects of the microtubule cytoskeleton. This reviewer cannot comment on whether researchers outside this area would find the findings interesting.

The approach seems scientifically sound, and the data quality is good and supports the major observations/conclusions of the paper. No major flaws that should prohibit publication have been identified.

This reviewer recognises that it would be difficult to perform functional knock-out experiments to target the new densities identified in sperm axonemes. Although it is important to validate and/or determine the identity of the novel structures (the barrel-shaped density and the connections between radial spokes) these experiments are likely beyond the scope of this study. Instead, the authors could attempt template matching approaches or structural homology-based searches of the ciliary proteome to identify the barrel shaped structures. There are limitations in doing this and this approach may not immediately yield additional insights but it would greatly add to the novelty of the study.

The averaging of repetitive units is inherently quantitative in nature and there are no additional quantitative experiments to suggest. The authors make useful comparisons using existing structural models of axonemes from other motile ciliated cell types and from several other species to emphasize the presence of the novel structures specific to mammalian sperm which they propose contribute to asymmetric beating.

Overall, the manuscript is well written barring minor grammatical and typographical errors which should be corrected upon further proof-reading (specific example: line 36 – eukaryotic flagella and motile cilia are...). Please use the appropriate plural form when referring to multiple flagella and cilia, there are several other instances where the usage of appropriate grammar will ensure that the reader is not distracted from the flow of the manuscript which is of an otherwise high quality.

The figures are well presented and easy to interpret. The figures look to have all been rendered entirely using UCSF ChimeraX, but the methods section states the use of UCSF Chimera. Please clarify and use the appropriate reference for UCSF ChimeraX. The length is justified to convey the findings of the study in a clear and direct manner.

On the basis of the abovementioned justifications, this reviewer recommends this study to be accepted for publication with the changes detailed above.

Minor comments

Lines 111 and 116

The authors do not speculate what could make up the extra densities? could there be additional Tektin-binding proteins? It is not satisfactory to simply make an observation without providing a possible explanation or insight.

Extended data figure 2

Although not as extensive as the cross-connecting A-tubule MIPs, there are additional densities in the B-tubule lumen attached to the inner wall of the lattice in mouse sperm which are conspicuously absent in bovine trachea axonemes. In the interest of being complete, it would be good to mention these densities.

Extended data figure 3 b

In the top panel in 3b, there is a clear density at the base of one of the IDAs which is coloured in pink. Do the authors ascribe this density to be part of one of the IDAs in mouse sperm? if so, this is quite a major structure which is missing in sea urchin sperm IDAs. Alternatively, could the authors have mis-coloured the barrel shaped density in pink? please clarify this in the colouring scheme.

Reviewer #3:

Remarks to the Author:

The manuscript uses cryo electron tomography and subtomogram averaging in the context of human and mouse sperm cells to show novel protein densities in mouse / human flagella in comparison to other species commonly used to study cilia architecture. The authors suggest that those new densities are in the right locations to be probably important to explain the molecular basis of the asymmetric flagellar beat and the higher mechanical strength of the mouse and human sperm flagella.

The finding of those novel protein densities in mouse and human sperm flagella is convincing and clearly interesting, therefore it surely advances the knowledge of the field.

The results shown in the figures and their interpretation seem robust and careful. In general, the manuscript is well written and clearly presented (including the abstract).

The data is of good quality and technically sound, but some processing aspects have to be improved and disclosed (see below).

The conclusions of the manuscript have a broad interest, but, as the authors admit in the discussion, there is a lack of evidence of the molecular identity of the new protein densities and if those densities are really critical in regulating the beat of mouse and human sperms. In general, the structural novelties of this paper are therefore not supported by other data. Frankly, it would be very difficult for

the authors to provide such experimental data in a reasonable time frame, but a list of the plausible proteins would make the manuscript more complete.

Important points to address:

1) The authors claim several resolution numbers, but they do not show the corresponding FSCs. The authors should show all the FSCs and for the main averages also the local resolutions.

2) According to Leung et al 2021 the outer dense fibers are important in the flagellar asymmetry since they are coupled to the doublets in the principal piece, but not in the midpiece. The authors should average and compare the positions of the outer dense fibers in mouse and human flagella to complete their structural work.

3) The method section is not very detailed while a lot of computational work has been performed. The authors should deposit the scripts that allowed the key discoveries of the paper and explain better the processing strategy in the methods. Here are the missing points:

- scripts to separate the different doublets for processing and mark the coordinates of subtomograms
- add a scheme of the processing strategy with Relion including the masks for the multibody refinement
- add details of how repicking and recentering were performed in the methods
- how were the maps segmented and coloured for figure preparation? Add details
- how many FIB lamellae were used for the mouse sperm dataset? Add this information
- show an exemplary tomographic slice for a human sperm tomogram to understand the quality of this dataset
- I would like to ask the authors to measure all the tomogram thicknesses for both mouse and human tomograms and plot it to show the difference between milled and unmilled tomos

4) The authors should deposit in the EMDB also the maps of the human flagella since they are present in many figures and an integral part of the results. In addition, since the quality of the mouse data is higher, it would help the community of developers to deposit the raw tilt series of the mouse flagella in the EMPIAR archive.

5) Could the author provide a list of flagellar associated proteins in mouse sperm? Such information would make the story rounder, because the new densities may belong to some of those names. The authors could perform a proteomic experiment or a list may be available in literature.

Minor points

1) In Extended data Fig4 a coloured segmentation instead of grey densities would be nice to help the reader, as done in the other figures

2) How many sperm cells per unity of volume were applied to the grid? (add in the methods the number for mouse and human)

Author Rebuttal to Initial comments

Point-by-point response

Reviewer #1:

Major points:

Line 180-182, Fig. 2d,e: This reviewer is not sure whether the averaging was done properly to derive the conclusion (doublets longitudinally not aligned each other). Since CP and doublets have 32nm and 96nm periodicity, if the alignment/averaging was done using CP, doublets, even if they are aligned, will be averaged out between the three (=96nm/32nm) possible shifts. To make sure if the doublets are aligned each other, they must be aligned as the whole axoneme (9+2) and averaged. Probably their conclusion is correct in the end, but from Fig. 2 it cannot be concluded so.

We agree with the reviewer and emphasized this point in the concluding sentence of the paragraph.

“Such lack of alignment could be caused by mismatch of the 32-nm periodicity of the central pair complex and the 96-nm periodicity of the doublet microtubules (Fig. 2d,e).”

Line 238-241, Extended Data Fig. 7: indexing of single-headed inner dyneins (after Chlamydomonas) is wrong. It should be from the proximal end dynein a, b, c, e, g, d (they indexed a, b, c, d, e, g). Please refer Bui et al. 2012 JCB.

We thank the reviewer for pointing out this error and we have updated the figure and text.

“The densities of inner arm dyneins were also generally similar for all doublets, with two exceptions. For doublet 5, densities corresponding to dynein e and g (nomenclature defined in *Chlamydomonas*¹⁵) were shifted compared to the other doublets”

Minor points:

Line 60-61: Bridge like structures between two adjacent doublets, which do not follow 9-fold symmetry, were pointed out in Chlamydomonas (Bui et al. 2009 JCB).

We apologize for missing the key reference and added it in the updated manuscript.

“A unique bridge-like structure that crosslinks two neighboring doublets is proposed to constrain the plane of bending¹⁵⁻¹⁷.”

Line114: “Protrusions extending from filamentous structure”: Please indicate them in Extended Data Fig. 2a.

We labeled the protrusions and the filamentous core by arrowheads and ovals in the updated Extended Data Fig. 3a (note the figure number changed).

Line140-142: Unique structure of human RS heads: does this structure have correlation to the difference between the authors structure of reconstituted RS head and Brown’s structure?

In order to improve the clarity, we added a figure panel to compare radial spokes of mouse axonemes to the ones from *Chlamydomonas* and *Tetrahymena* (Extended Data Fig. 4c) and rewrote the original sentences with references.

“These radial spoke features are similar to the ones observed in human respiratory cilia²⁷. In contrast, the heads of RS1 and RS2 from *Chlamydomonas* and *Tetrahymena* do not have the deep cleft^{11-13,25,26}. The RS3 stump from *Chlamydomonas* is much shorter and RS3 from *Tetrahymena* has a smaller surface area of the head^{13,25} (Extended Data Fig. 4c)”

Line177: “for the center central pair complex”: it is not clear.

We thank the reviewer for pointing out this typo as we missed an “of” in the original sentence. We have now corrected it:

“Although the alignment was only performed for the center of central pair complex in the expanded subvolumes, nine distinct doublet microtubule densities that are parallel to the singlet microtubules could be resolved, indicating a remarkably consistent radial arrangement of the nine doublets in the axonemes.”.

Line209-213: This reviewer could not follow the description. Indications in Fig.3d will help.

We re-drew the Figure 3 to better show the complementary shapes of central pair protrusions and the radial spokes. We then concluded the paragraph with just the following sentence:

“Such complementary shapes may limit the movement of doublet 8 sideways and stabilize its radial position.”

Most of the previous discussion was simplified and moved in the Discussion section in reference to Fig. 7c.

“This idea is consistent with the sliding hypothesis for axonemal bending, in which active dyneins generate displacement between the neighboring doublets³⁹. Such movement would also lead to displacement of the doublets relative to the central pair complex along the longitudinal axis of axonemes, as if there are nine trains (doublets) move along nine tracks (central pair protrusions) (Fig. 7c).”

Line 394, Fig. 7d: It will be helpful if it is shown how much displacement should happen in the absence of 5-6 linker with normal axoneme curvature during the beating.

We added the following discussion on estimation of offsets based on reported bending curvature of mouse sperm flagella in the Discussion section.

“As an estimate based on the reported curvature of mouse sperm³², the offsets between neighboring doublets could differ as much as $\Delta \sim 28$ nm within one tomogram (See details in the Methods section). More importantly, the initial offset of each tomogram varies depending on how much sliding has happened upstream or downstream of the imaged area of the flagella. In contrast, the resolved periodic features in both doublet 5 and doublet 6 at the same time suggest there is a consistent offset between these two doublets, not just within each tomogram but also among the 63 tomograms that contribute to the average. Considering the nature of random sampling of our imaging areas of different cells ($N = 63$ tomograms), this could happen if there is limited bending along the direction parallel to the plane of these two filaments in mouse flagella (Fig. 7e).”

We added the deduction of offsets in the Methods section:

“The bending curvature of mouse sperm could be as big as $2 \times 10^5 \text{ m}^{-1}$ or $0.20 \mu\text{m}^{-1}$ based on literature ($\text{OD} = 5 \mu\text{m} = 5000 \text{ nm}$)¹². The distance between the neighboring doublets is 72 nm (measured in the average of the axoneme shown in Fig. 2d, $\text{OB} = 5072 \text{ nm}$). If AB and CD represent one 96-nm repeat on Doublet 1 and Doublet 2, respectively, offset $\Delta \sim \text{CE} = \text{OC} \times (\angle \text{COD} - \angle \text{AOB}) = 5000 \times (96/5000 - 96/5072) = 1.4 \text{ nm}$.

Note the Δ represents the additional offset per 96 nm repeat. In a tomogram that contains an axoneme of $\sim 2 \mu\text{m}$, if the offset between the first pair of 96-nm repeats is 0 nm, the offset of the 20th pair is 28 nm. Note this initial offset of the first pair (0 nm) is tomogram-specific depending on how much sliding has happened upstream or downstream of the imaging area and the imaged cell.”

Fig.7c: What are red arrows?

We edited the legend for Fig. 7c to explain the arrows.

“Our multibody analyses are consistent with the sliding hypothesis and potential longitudinal movements of dmt1 are indicated by the two arrows, like “a train on a track”.”

Line577: This should be for panel g.

We thank the reviewer for pointing this typographic error out and we have corrected it.

Methods: How principal component analysis of subtomograms was done should be written.

We added the following paragraph in the Methods section to better explain how the Principal Component Analysis/Multi-body Refinement was done and also Extended Data Fig. 2c to illustrate the Multi-body Refinement workflow:

“For the multi-body refinement (see the schematic workflow in Extended Data Fig. 2c), the subvolumes corresponding to specific radial spoke-central pair interface were re-extracted based on the subvolumes of 96-nm units of individual doublet microtubules. The particles were recentered at the junction between the head and the stalk of radial spoke 2 to include enough features from both the doublets and central pair complex. The radial spokes were then aligned with a mask. This mask and a mask covering the central pair complex were used for the multibody analysis implemented in RELION3⁸. Briefly, separate refinement of the radial spokes and central pair protrusions provided two sets of alignment parameters: three translational shifts (x,y,z) and three Euler angles required to align one subvolume to each reference. Thus 12 parameters can be used to remap the two references back to each raw subvolume and their spatial relationship in each subvolume could be determined by 12 parameters. Principal Component Analyses (PCA) essentially reprojected the original 12-dimensional data in a new 12-dimensional space with 12 new orthogonal eigenvectors. These 12 eigenvectors could be mathematically determined so that they represent decreasing variations of the entire data along the individual eigenvectors. Our analyses suggest the first eigenvector/axis is parallel to the axoneme axis and the variations along this axis account for 40-50% of the total variations. When all subvolumes were divided into 10 groups based on their projection values on the

first axis (10% lowest, 10%-20% lowest, ..., 90%-100% highest). Each group was then represented by a snapshot depicting the averaging projection values of the group members and 10 of these snapshots were morphed to generate the movie (Supplementary Movie 1).”

Reviewer #2:

Major comments

Specific to the findings on per-doublet averaging and connections to the central pair, this reviewer would point the authors to the following preprint 'Cryo-EM structure of an active central apparatus' DOI: <https://doi.org/10.1101/2022.01.23.477438>. This preprint reports repeating surface densities on high resolution models of the central pair apparatus very similar to the one's highlighted in figure 2 and extended data figure 5 of the study under review. The authors of the current study could benefit from a more detailed comparison with the structural models presented in the suggested preprint. Such a comparison could reveal unique features specific to mammalian sperm central pairs.

We thank the review's recommendations. We did look into both of the recently published papers on the cryoEM structures of the central pair complex from *Chlamydomonas* in *Nat. Struct. Mol. Biol* and we will keep these high-res structures in mind moving forward. Currently, we are trying to improve the resolution of our maps to identify the molecular differences for the future studies, but this is out of the scope of the current work.

The central pair complex of mammalian sperm is very similar to the one from sea urchin sperm. At slightly lower resolutions (33-40 Å), the comparison of the central pair complex of sea urchin sperm and the one from *Chlamydomonas* has been done (Carbajal-Gonzalez, et al. *Cytoskeleton* 2013) and they were found to be quite different. We thus edited the original sentence in the Results section to clarify that:

“Notably, compared to the central pair complex of sea urchin sperm, where MIPs were not observed, the overall shapes of the external protrusions are very similar, while the central pair complex from sea urchin sperm was found to be different to that of *Chlamydomonas* flagella²³. These comparisons suggest that the asymmetric surface of the central pair complex is likely conserved from invertebrate to vertebrate sperm (animal sperm), but different from the ones from unicellular protists (Extended Data Fig. 5b,c).”

This reviewer recognises that it would be difficult to perform functional knock-out experiments to target the new densities identified in sperm axonemes. Although it is important to validate and/or determine the identity of the novel structures (the barrel-shaped density and the connections between radial spokes) these experiments are likely

beyond the scope of this study. Instead, the authors could attempt template matching approaches or structural homology-based searches of the ciliary proteome to identify the barrel shaped structures. There are limitations in doing this and this approach may not immediately yield additional insights but it would greatly add to the novelty of the study.

We agree with the reviewers #2 and #3 that identifying the sperm-specific proteins is the next key step. Based on our current resolutions we feel that this is beyond the scope of this manuscript. Currently, the published template-matching methods used single micrographs at 0° (by Grigorieff lab) instead of tilt series so it would require a new dataset and methodology development for sperm-related proteins that are much smaller than the ribosomes studied in the published work. On the other hand, the homology-search method requires subnanometer resolutions for the subtomogram averages, which we are working towards. We do however appreciate these reviewers' suggestions, and look forward to working with these ideas and finding out the molecular identities of sperm-specific densities.

Overall, the manuscript is well written barring minor grammatical and typographical errors which should be corrected upon further proof-reading (specific example: line 36 – eukaryotic flagella and motile cilia are...). Please use the appropriate plural form when referring to multiple flagella and cilia, there are several other instances where the usage of appropriate grammar will ensure that the reader is not distracted from the flow of the manuscript which is of an otherwise high quality.

We thank the reviewer for pointing out the error and we have corrected it.

The figures are well presented and easy to interpret. The figures look to have all been rendered entirely using UCSF ChimeraX, but the methods section states the use of UCSF Chimera. Please clarify and use the appropriate reference for UCSF ChimeraX. The length is justified to convey the findings of the study in a clear and direct manner.

We thank the reviewer for pointing out the error. We added UCSF Chimera X in the description and two references.

Minor comments

Lines 111 and 116

The authors do not speculate what could make up the extra densities? could there be additional Tektin-binding proteins? It is not satisfactory to simply make an observation without providing a possible explanation or insight.

Currently, we do not know other Tektin-binding molecules in addition to previously identified Tektin (Gui et al 2021). In terms of identifying potential candidate proteins for extra densities, we added the following discussion on a sperm-specific Tektin-5.

“Tektin-1 to -4 have been known to assemble into three-helix bundles that pack along one another laterally inside the microtubule doublets of bovine trachea cilia²⁴, and likely compose much of the filamentous densities observed in sperm. Sperm also contain an additional tektin (Tektin-5; see Supplementary Data 2) which is a candidate for some of the additional sperm-specific densities (as shown in Extended Data Fig. 3), although the precise assignment requires higher-resolution reconstructions.”

Extended data figure 2

Although not as extensive as the cross-connecting A-tubule MIPs, there are additional densities in the B-tubule lumen attached to the inner wall of the lattice in mouse sperm which are conspicuously absent in bovine trachea axonemes. In the interest of being complete, it would be good to mention these densities.

The reviewer brought up a great point. We added the Extended Data Fig. 3c and also labels in Extended Data Fig. 3b to highlight the MIPs in the B-tubule and the following statement in the Results section:

“In addition, we observed striated densities along the helical pitch of microtubule inside the B-tubule. These striations were separated by 8 nm and cover the intradimeric interface between the α - and β -tubulins (Extended Data Fig. 2b,c).”

We then added a sentence in the Discussion section on their possible roles in strengthening the microtubule filaments.

“Also, the proteins that form the striations inside the B-tubule could crosslink the tubulins within B2-B6 protofilaments while also couple these protofilaments laterally along the helical pitch.”

Extended data figure 3 b

In the top panel in 3b, there is a clear density at the base of one of the IDAs which is coloured in pink. Do the authors ascribe this density to be part of one of the IDAs in mouse sperm? if so, this is quite a major structure which is missing in sea urchin sperm IDAs. Alternatively, could the authors have mis-coloured the barrel shaped density in pink? please clarify this in the colouring scheme.

The reviewer is correct that those densities are from the radial spoke barrel. We added labels in Extended Data Figure 4b to clarify that.

“Note the barrel densities between RS1 and RS2 only exist in the mouse sperm are indicated (dashed ovals).”

Reviewer #3:

Remarks to the Author:

The conclusions of the manuscript have a broad interest, but, as the authors admit in the discussion, there is a lack of evidence of the molecular identity of the new protein densities and if those densities are really critical in regulating the beat of mouse and human sperms. In general, the structural novelties of this paper are therefore not supported by other data. Frankly, it would be very difficult for the authors to provide such experimental data in a reasonable time frame, but a list of the plausible proteins would make the manuscript more complete.

Important points to address:

1) The authors claim several resolution numbers, but they do not show the corresponding FSCs. The authors should show all the FSCs and for the main averages also the local resolutions.

We have added Supplementary Data 1 that includes 24 FSCs for the density maps shown in the figures. These maps include the consensus averages of doublets and the central pair complex, as well as the per-doublet averages (doublet 1-9) of both human and mouse sperm, plus the 48 nm-repeating structure of the MIPs and the 5-6 bridge of mouse sperm. We also labeled the FSC curves with the EMDB numbers and resolution estimate (FSC = 0.143). All 24 EMDB entries are submitted and held for release upon publication.

We have added Supplementary Data 2 that illustrates the local resolution maps for the consensus averages of doublets and the central pair complex of both human and mouse sperm. These were generated by blocres from the Bsoft package. We also noted in the Methods section:

“Local resolution maps for the consensus averages of doublets and the central pair complex of both human and mouse sperm were calculated by blocres in Bsoft (Supplementary Data 2). These local-resolution maps represent relative differences in resolution across the maps but the absolute values may not be exact.”

2) According to Leung et al 2021 the outer dense fibers are important in the flagellar asymmetry since they are coupled to the doublets in the principal piece, but not in the midpiece. The authors should average and compare the positions of the outer dense fibers in mouse and human flagella to complete their structural work.

We also looked at the outer dense fibers during data analysis. However, our averages do not show densities connecting the outer dense fibers to the doublets. We have added Extended Data Fig. 8b to highlight the specific and complex connections between the nine doublets and their corresponding outer dense fibers, which suggest an even bigger dataset is required to address such questions appropriately. We noted in the Result section:

“In these per-doublet averages, we did not resolve robust densities connecting the outer dense fibers to the doublets. Previous studies suggest averages of all nine doublets in the proximal principal region have such attachments²¹. However, our raw tomograms of the proximal principal piece showed that some outer dense fibers are close to the corresponding doublets and some are further away (Extended Data Fig. 8b). Such variations require per-doublet averages to be considered. However, the subset of tomograms in the proximal principal is small and per-doublet averaging resulted in anisotropic 3D reconstructions. An even larger dataset is required to resolve structures with such a specific and complex distribution pattern.”

3) The method section is not very detailed while a lot of computational work has been performed.

The authors should deposit the scripts that allowed the key discoveries of the paper and explain better the processing strategy in the methods. Here are the missing points:

We added more comprehensive information as the reviewer requested (see point-by-point details below).

- scripts to separate the different doublets for processing and mark the coordinates of subtomograms

We added the script in the Methods section with a header: “Script to remap coordinates of subvolumes in three dimensions”.

- add a scheme of the processing strategy with Relion including the masks for the multibody refinement

We added Extended Data Fig. 2a,b to illustrate the processing schemes using RELION3 and also Extended Data Fig. 2c for the multi-body refinement showing the masks. In addition, we added a detailed description for the latter in the Methods section as requested by Reviewer#1.

“For the multi-body refinement (see the schematic workflow in Extended Data Fig. 2c), the subvolumes corresponding to specific radial spoke-central pair interface were re-extracted based on the subvolumes of 96-nm units of individual doublet microtubules. The particles were recentered at the junction between the head and the stalk of radial spoke 2 to include enough features from both the doublets and central pair complex. The radial spokes were then aligned with a mask. This mask and a mask covering the central pair complex were used for the multibody analysis implemented in RELION3⁸. Briefly, separate refinement of the radial spokes and central pair protrusions provided two sets of alignment parameters: three translational shifts (x,y,z) and three Euler angles required to align one subvolume to each reference. Thus 12 parameters can be used to remap the two references back to each raw subvolume and their spatial relationship in each subvolume could be determined by 12 parameters. Principal Component Analyses (PCA) essentially reprojected the original 12-dimensional data in a new 12-dimensional space with 12 new orthogonal eigenvectors. These 12 eigenvectors could be mathematically determined so that they represent decreasing variations of the entire data along the individual eigenvectors. Our analyses suggest the first eigenvector/axis is parallel to the axoneme axis and the variations along this axis account for 40-50% of the total variations. When all subvolumes were divided into 10 groups based on their projection values on the first axis (10% lowest, 10%-20% lowest, ..., 90%-100% highest). Each group was then represented by a snapshot depicting the averaging projection values of the group members and 10 of these snapshots were morphed to generate the movie (Supplementary Movie 1).”

- add details of how repicking and recentering were performed in the methods

We added Extended Data Fig. 2a,b to show how repicking and recentering were performed. In addition, we added the following statement in the Methods section:

“The Pixel View in IMOD was used to determine the coordinates corresponding to the base of radial spoke 2 in all four class averages and all subvolumes were re-extracted in RELION and recentered to that same point.”

- how were the maps segmented and coloured for figure preparation? Add details

We added “UCSF Chimera was used to manually segment the maps for various structure features and these maps were colored individually to prepare the figures.” in the Methods section.

- how many FIB lamellae were used for the mouse sperm dataset? Add this information

We added a paragraph in the Methods section detailing the information about both the milling data of mouse sperm and non-milling data of human sperm.

“In total, we started with 24 milling grids of mouse sperm and obtained 200 lamellae. Tilt series with no crystal ice were kept and further processed. In some cases, a significant part of the (9+2) axoneme is milled away and we only processed the ones with at least 5 doublets and also enough space including the central pair complex for subvolume averaging. In the end, the final reconstructions of consensus averages were from 70 usable tomograms. For the human sperm cryoET dataset, the final reconstructions of consensus averages were from 65 tilt series with the flagella orientations within $\sim 30^\circ$ of the tilt axis. Note the final tomograms used for different reconstructions vary as tomograms may be discarded due to bad alignment or may lack certain doublets.”

- show an exemplary tomographic slice for a human sperm tomogram to understand the quality of this dataset

We added figure panels in Extended Data Fig. 9a-b to show the xy and xz slices of a representative human sperm tomogram and noted in the legend “Although the A- and B-tubule of the microtubule doublets are discernible, other features like axonemal dyneins and radial spokes are not clear in this slice. This is in contrast to the example of mouse sperm lamella shown in Extended Data Fig. 1h-i”.

- I would like to ask the authors to measure all the tomogram thicknesses for both mouse and human tomograms and plot it to show the difference between milled and unmilled tomos

We added Extended Data Fig. 9c-e showing the estimated thickness of lamellae and noted in the legend on how we measured them. For the human sperm data, as shown in the Extended Fig. 9d, the top and bottom boundaries of the cell are not resolved due to the missing wedge so we measured the thickness of nearby ice instead. The measured value represents a lower bound of the actual thickness of human sperm (also noted in the legend).

4) The authors should deposit in the EMDB also the maps of the human flagella since

they are present in many figures and an integral part of the results. In addition, since the quality of the mouse data is higher, it would help the community of developers to deposit the raw tilt series of the mouse flagella in the EMPIAR archive.

We deposited the following 24 maps in EMD. These entries include the consensus averages of doublets and the central pair complex, as well as the per-doublet averages (doublet 1-9) from both human and mouse sperm, as well as the 48 nm-repeating structure of the MIPs and the 5-6 bridge from the mouse sperm. In the Data Availability section, we noted:

“The maps of the following structures are available in the Electron Microscopy Data Bank: EMD-27444, consensus average of 96 nm-repeating structure of mouse doublets; EMD-27445, 32 nm-repeating structure of central pair complex of mouse sperm; EMD-27446, mouse doublet 1; EMD-27447, mouse doublet 2; EMD-27448, mouse doublet 3; EMD-27449, mouse doublet 4; EMD-27450, mouse doublet 5; EMD-27451, mouse doublet 6; EMD-27452, mouse doublet 7; EMD-27453, mouse doublet 8; EMD-27454, mouse doublet 9; EMD-27455, 48 nm-repeating structure of doublet microtubule of mouse sperm; EMD-27456, 5-6 bridge of mouse sperm; EMD-27462, consensus average of 96 nm-repeating structure of human doublets; EMD-27463, 32 nm-repeating structure of central pair complex of human sperm; EMD-27464, human doublet 1; EMD-27465, human doublet 2; EMD-27466, human doublet 3; EMD-27467, human doublet 4; EMD-27468, human doublet 5; EMD-27469, human doublet 6; EMD-27470, human doublet 7; EMD-27471, human doublet 8; EMD-27473, human doublet 9. The raw tilt series of mouse sperm lamellae are available in the EMPIAR database.”

We have prepared the raw tilt series of milling mouse flagella and they are ready to be deposited in the EMPAIR archive. However, the database asked for citation and journal information that would only be available later. We are happy to deposit the data once we obtain the relevant information.

5) Could the author provide a list of flagellar associated proteins in mouse sperm? Such information would make the story rounder, because the new densities may belong to some of those names. The authors could perform a proteomic experiment or a list may be available in literature.

We thank the reviewer's suggestion and annotated a published mass spectrometry studies of bovine sperm in the Supplementary Table S1. In the Discussion section, we postulated that Tektin-5 could contribute to part of the sperm-specific MIP densities:

“Tektin 1-4 have been known to assemble into three-helix bundles that pack along one another laterally inside the microtubule doublets of bovine trachea cilia²⁴, and likely

compose much of the filamentous densities observed in sperm. Sperm also contain an additional tektin (Tektin-5; see Supplementary Data 2) which is a candidate for the additional sperm-specific density (as shown in Extended Data Fig. 3), although this assignment requires confirmation with high resolution.”

Minor points

1) In Extended data Fig4 a coloured segmentation instead of grey densities would be nice to help the reader, as done in the other figures

We updated the Extended Data Fig. 4 with the same color scheme as Figure 1.

2) How many sperm cells per unity of volume were applied to the grid? (add in the methods the number for mouse and human)

We added the following statement in the Methods section:

“The mouse sperm suspension was then mixed with 10-nm gold beads (Electron Microscopy Science, cat #25487) to achieve final concentrations at $2-6 \times 10^6$ million/mL. The human sperm suspension was mixed with 10-nm gold beads to achieve final concentrations at $0.5-2 \times 10^6$ million/mL.”

Decision Letter, first revision:

Our ref: NSMB-A46123A

12th Sep 2022

Dear Dr. Vale,

Thank you for submitting your revised manuscript "In situ cryo-electron tomography reveals the asymmetric architecture of mammalian sperm axonemes" (NSMB-A46123A). It has now been seen by the original referees and their comments are below. The reviewers find that the paper has improved in revision, and therefore we'll be happy in principle to publish it in Nature Structural & Molecular Biology, pending minor revisions to comply with our editorial and formatting guidelines.

Kind regards,
Florian

Dr Florian Ullrich
Associate Editor, Nature
Consulting Editor, Nature Structural & Molecular Biology
ORCID 0000-0002-1153-2040

Reviewer #1 (Remarks to the Author):

The authors addressed all the points raised by this reviewer. This reviewer thinks points by other reviewers were also addressed well. This manuscript is ready for publication. Only one small concern is about point #1 by reviewer #2. Extended Fig.5BC compares sea urchin and mammalian sperm flagella structures, but not with unicellular axoneme (such as Chlamydomonas). Therefore it is not appropriate to state "...but different from the ones from unicellular protists (Extended Data Fig. 5b,c)". Comparison with recently published Chlamydomonas CP structure might be still necessary.

Reviewer #2 (Remarks to the Author):

Novel finding of asymmetrically positioned barrel-like structures plus additional densities.

Insightful application of sub-tomogram averaging to explain the molecular basis of a biological phenomenon (asymmetric beat of sperm flagella in mammals).

Good data quality and presentation.

Took reviewer comments on board and addressed concerns satisfactorily.

Reviewer #3 (Remarks to the Author):

I would like to congratulate the authors for the excellent rebuttal, new text and figures. All my points have been addressed carefully and fully! The high-quality data coupled with the extremely well-designed strategies for data analysis allow brilliantly the novel conclusions. I see this article as a significant advance in the understanding of sperm flagellar architecture and beating function, hence I strongly support its publication in NSMB.

Decision Letter, Final Checks:

Our ref: NSMB-A46123A

15th Sep 2022

Dear Dr. Vale,

Thank you for your patience as we've prepared the guidelines for final submission of your Nature Structural & Molecular Biology manuscript, "In situ cryo-electron tomography reveals the asymmetric architecture of mammalian sperm axonemes" (NSMB-A46123A). Please carefully follow the step-by-step instructions provided in the attached file, and add a response in each row of the table to indicate the changes that you have made. Please also check and comment on any additional marked-up edits we have proposed within the text. Ensuring that each point is addressed will help to ensure that your revised manuscript can be swiftly handed over to our production team.

In recognition of the time and expertise our reviewers provide to Nature Structural & Molecular Biology's editorial process, we would like to formally acknowledge their contribution to the external peer review of your manuscript entitled "In situ cryo-electron tomography reveals the asymmetric architecture of mammalian sperm axonemes". For those reviewers who give their assent, we will be publishing their names alongside the published article.

Nature Structural & Molecular Biology offers a Transparent Peer Review option for new original research manuscripts submitted after December 1st, 2019. As part of this initiative, we encourage our authors to support increased transparency into the peer review process by agreeing to have the reviewer comments, author rebuttal letters, and editorial decision letters published as a Supplementary item. When you submit your final files please clearly state in your cover letter whether or not you would like to participate in this initiative. Please note that failure to state your preference will result in delays in accepting your manuscript for publication.

Cover suggestions

As you prepare your final files we encourage you to consider whether you have any images or illustrations that may be appropriate for use on the cover of Nature Structural & Molecular Biology.

Nature Structural & Molecular Biology has now transitioned to a unified Rights Collection system which will allow our Author Services team to quickly and easily collect the rights and permissions required to publish your work. Approximately 10 days after your paper is formally accepted, you will receive an email in providing you with a link to complete the grant of rights. If your paper is eligible for Open Access, our Author Services team will also be in touch regarding any additional information that may be required to arrange payment for your article.

Please note that *Nature Structural & Molecular Biology* is a Transformative Journal (TJ). Authors may publish their research with us through the traditional subscription access route or make their paper immediately open access through payment of an article-processing charge (APC). Authors will not be required to make a final decision about access to their article until it has been accepted. [Find out more about Transformative Journals](https://www.springernature.com/gp/open-research/transformative-journals)

Authors may need to take specific actions to achieve [compliance with funder and institutional open access mandates](https://www.springernature.com/gp/open-research/funding/policy-compliance-faqs). If your research is supported by a funder that requires immediate open access (e.g. according to [Plan S principles](https://www.springernature.com/gp/open-research/plan-s-compliance)) then you should select the gold OA route, and we will direct you to the compliant route where possible. For authors selecting the subscription publication route, the journal's standard licensing terms will need to be accepted, including [self-archiving policies](https://www.nature.com/nature-portfolio/editorial-policies/self-archiving-and-license-to-publish). Those licensing terms will supersede any other terms that the author or any third party may assert apply to any version of the manuscript.

Please use the following link for uploading these materials:
[Redacted]

Best regards,

Sophia Frank
Editorial Assistant
Nature Structural & Molecular Biology
nsmb@us.nature.com

On behalf of

Florian Ullrich, Ph.D.
Associate Editor
Nature Structural & Molecular Biology
ORCID 0000-0002-1153-2040

Reviewer #1:

Remarks to the Author:

The authors addressed all the points raised by this reviewer. This reviewer thinks points by other reviewers were also addressed well. This manuscript is ready for publication. Only one small concern is about point #1 by reviewer #2. Extended Fig.5BC compares sea urchin and mammalian sperm flagella structures, but not with unicellular axoneme (such as Chlamydomonas). Therefore it is not appropriate to state "...but different from the ones from unicellular protists (Extended Data Fig. 5b,c)". Comparison with recently published Chlamydomonas CP structure might be still necessary.

Reviewer #2:

Remarks to the Author:

Novel finding of asymmetrically positioned barrel-like structures plus additional densities.

Insightful application of sub-tomogram averaging to explain the molecular basis of a biological phenomenon (asymmetric beat of sperm flagella in mammals).

Good data quality and presentation.

Took reviewer comments on board and addressed concerns satisfactorily.

Reviewer #3:

Remarks to the Author:

I would like to congratulate the authors for the excellent rebuttal, new text and figures. All my points have been addressed carefully and fully! The high-quality data coupled with the extremely well-designed strategies for data analysis allow brilliantly the novel conclusions. I see this article as a significant advance in the understanding of sperm flagellar architecture and beating function, hence I strongly support its publication in NSMB.

Author Rebuttal, first revision:

Point-by-point response

Reviewer #1:

Remarks to the Author:

*The authors addressed all the points raised by this reviewer. This reviewer thinks points by other reviewers were also addressed well. This manuscript is ready for publication. Only one small concern is about point #1 by reviewer #2. Extended Fig.5BC compares sea urchin and mammalian sperm flagella structures, but not with unicellular axoneme (such as *Chlamydomonas*). Therefore it is not appropriate to state "...but different from the ones from unicellular protists (Extended Data Fig. 5b,c)". Comparison with recently published *Chlamydomonas* CP structure might be still necessary.*

We now added Extended Data Fig. 5c to directly compare the central pair complex of mouse sperm to the published map of *Chlamydomonas* central pair complex to highlight the differences (instead of just referring to reference of comparing Sea urchin sperm with *Chlamydomonas* structures).

Reviewer #2:

Remarks to the Author:

Novel finding of asymmetrically positioned barrel-like structures plus additional densities.

Insightful application of sub-tomogram averaging to explain the molecular basis of a biological phenomenon (asymmetric beat of sperm flagella in mammals).

Good data quality and presentation.

Took reviewer comments on board and addressed concerns satisfactorily.

We thank the reviewer for the kind and thoughtful comments.

Reviewer #3:

Remarks to the Author:

I would like to congratulate the authors for the excellent rebuttal, new text and figures. All my points have been addressed carefully and fully! The high-quality data coupled with the extremely well-designed strategies for data analysis allow brilliantly the novel conclusions. I see this article as a significant advance in the understanding of sperm flagellar architecture and beating function, hence I strongly support its publication in NSMB.

We thank the reviewer for the kind and thoughtful comments.

Final Decision Letter:

11th Oct 2022

Dear Dr. Vale,

We are now happy to accept your revised paper "In situ cryo-electron tomography reveals the asymmetric architecture of mammalian sperm axonemes" for publication as a Article in Nature Structural & Molecular Biology.

Due to the importance of these deadlines, we ask that you please let us know now whether you will be difficult to contact over the next month. If this is the case, we ask you provide us with the contact

information (email, phone and fax) of someone who will be able to check the proofs on your behalf, and who will be available to address any last-minute problems.

As soon as your article is published, you can generate your shareable link by entering the DOI of your article here: http://authors.springernature.com/share.

Corresponding authors will also receive an automated email with the shareable link

Note the policy of the journal on data deposition:

<http://www.nature.com/authors/policies/availability.html>.

Your paper will be published online soon after we receive proof corrections and will appear in print in the next available issue. You can find out your date of online publication by contacting the production team shortly after sending your proof corrections. Content is published online weekly on Mondays and Thursdays, and the embargo is set at 16:00 London time (GMT)/11:00 am US Eastern time (EST) on the day of publication. Now is the time to inform your Public Relations or Press Office about your paper, as they might be interested in promoting its publication. This will allow them time to prepare an accurate and satisfactory press release. Include your manuscript tracking number (NSMB-A46123B) and our journal name, which they will need when they contact our press office.

About one week before your paper is published online, we shall be distributing a press release to news organizations worldwide, which may very well include details of your work. We are happy for your institution or funding agency to prepare its own press release, but it must mention the embargo date and Nature Structural & Molecular Biology. If you or your Press Office have any enquiries in the meantime, please contact press@nature.com.

Please note that *Nature Structural & Molecular Biology* is a Transformative Journal (TJ). Authors may publish their research with us through the traditional subscription access route or make their paper immediately open access through payment of an article-processing charge (APC). Authors will not be required to make a final decision about access to their article until it has been accepted. Find out more about Transformative Journals

Authors may need to take specific actions to achieve compliance with funder and institutional open access mandates. If your research is supported by a funder that requires immediate open access (e.g. according to Plan S principles) then you should select the gold OA route, and we will direct you to the compliant route where possible. For authors selecting the subscription publication route, the journal's standard licensing terms will need to be accepted, including self-archiving policies. Those licensing terms will supersede any other terms that the author or any third party may assert apply to any version of the manuscript.

Kind regards,
Florian

Dr Florian Ullrich
Associate Editor, Nature
Consulting Editor, Nature Structural & Molecular Biology
ORCID 0000-0002-1153-2040